# Why Are Long-Read Sequencing Methods Revolutionizing Microbiome Analysis?

**DOI:** 10.3390/microorganisms13081861

**Published:** 2025-08-09

**Authors:** Adriana González, Asier Fullaondo, Adrian Odriozola

**Affiliations:** Department of Genetics, Physical Anthropology and Animal Physiology, University of the Basque Country (UPV/EHU), 48940 Bilbao, Spain; adriana.gonzalez@ehu.eus (A.G.); asier.fullaondo@ehu.eus (A.F.)

**Keywords:** microbiome, ecosystems, long-read sequencing, taxonomic resolution, variants, genome assembly, 16S, metagenomics

## Abstract

Most of the knowledge available on the composition and functionality of microbial communities in different ecosystems comes from short-read sequencing methods. It implies limitations regarding taxonomic resolution, variant detection, and genome assembly contiguity. Long-read sequencing technologies can overcome these limitations, transforming the analysis of microbial community composition and functionality. It is essential to understand the characteristics of each sequencing technology to select the most suitable one for each microbiome study. This review aims to show how long-read sequencing methods have revolutionized microbiome analysis in ecosystems and to provide a practical tool for selecting sequencing methods. To this end, the evolution of sequencing technologies, their advantages and disadvantages for microbiome studies, and the new dimensions enabled by long-read sequencing technologies, such as virome and epigenetic analysis, are described. Moreover, desirable characteristics for microbiome sequencing technologies are proposed, including a visual comparison of available sequencing platforms. Finally, amplicon and metagenomics approaches and the sequencing depth are discussed when using long-read sequencing technologies in microbiome studies. In conclusion, although no single sequencing method currently possesses all the ideal features for microbiome analysis in ecosystems, long-read sequencing technologies represent an advancement in key aspects, including longer read lengths, higher accuracy, shorter runtimes, higher output, more affordable costs, and greater portability. Therefore, more research using long-read sequencing is recommended to strengthen its application in microbiome analysis.

## 1. Introduction

In recent decades, short-read DNA sequencing methods have made it possible to explore the composition and functionality of microbiota in different ecosystems. The microbiota is the community of microorganisms, including archaea, bacteria, fungi, algae, and protists, that inhabit living organisms (e.g., human saliva, Asian elephant gut) or environmental settings (e.g., permafrost or the aquatic habitat of rainbow trout) [1,2,3,4]. These microorganisms play a central role in community assembly processes at the ecosystem level, ecosystem ecological functions (public health, productivity, nutrient cycling, and resistance to external disturbances), and ecosystem services (provisioning, regulating, cultural, and supporting) [5].

While short-read technologies have been essential in shedding light on key aspects of microbial ecosystems, they have important limitations regarding taxonomic resolution, variant detection, and genome reconstruction [6,7]. Long-read sequencing technologies, such as the sequencing platforms of Pacific Biosciences (PacBio) and Oxford Nanopore Technologies (ONT), have overcome many of these shortcomings, enabling a more accurate and detailed characterization of microorganisms present in specific environments, as well as in living organisms overall [6,7].

This technological advance is transforming the study of microbial biodiversity, the interactions between microbiota and their environment, and the functions within ecosystems, directly impacting human, animal, and environmental health [6,7].

Therefore, long-read sequencing enables complex microbiome challenges to be addressed from an ecosystem perspective, such as the Microbiome One Health Model. This model underlines the importance of understanding the structure and function of microbial communities in different ecosystems to address global challenges such as biodiversity loss, antimicrobial resistance, zoonosis, food security, and climate change [8]. The One Health concept has emerged as an integrative approach that recognizes the close link and interdependence between the health of humans, animals, plants, and the environment [9]. This concept is now a key element in the initiatives developed by the World Health Organization (WHO), highlighting the potential of long-read sequencing technologies to address global challenges [10].

Due to the increasing availability of long-read sequencing technologies and their transformative potential in ecosystem microbiome research, it is essential to understand their main features and applications. Although evidence regarding the role of sequencing technologies in microbiome studies is growing, the choice of the method best suited to each scientific objective is difficult because of its technical complexity and the scarcity of visual comparisons.

This review aims to explore the potential advantages of long-read sequencing in studies on the microbiomes of ecosystems by (1) analyzing the historical evolution of sequencing technologies and their advantages and disadvantages for microbiome analysis; (2) elaborating a proposal for desirable characteristics of microbiome sequencing methods to facilitate the choice according to the study’s objectives; and (3) providing an informative and visual comparison of the main platforms used in microbiome analysis. Finally, other important points are addressed, including amplicon and metagenomics approaches, as well as sequencing depth.

## 2. Evolution of Sequencing Technologies for Microbiome Analysis in Ecosystems

The following sections explore the evolution of sequencing technologies used to study microbiomes in ecosystems (Figure 1). The objective is to provide the reader with an overview of the currently most widely used sequencing for analyzing the composition and functionality of microbial communities and explaining the reasons for their choice and their main advantages and limitations.

## 3. The Early Development of Microbiome Sequencing: From Culture to First-Generation Sequencing

Before microbial sequencing, near-full-length 16S ribosomal RNA (hereinafter 16S) gene amplicons were cloned to isolate the target fragment and facilitate purification and storage [11]. The majority of microorganisms cannot be cultured in the laboratory. Therefore, shifting from culture to sequencing-based methods has allowed a more accurate sample characterization of microbial community diversity [12].

In the first generation of sequencing technologies, Sanger was used as the primary method for sequencing analysis and phylogenetic identification of microbial communities. Sanger sequencing is based on the random addition of chain terminators (dideoxynucleotides) during DNA replication. The dideoxy nucleotides contain no 3′-hydroxyl group; therefore, once DNA polymerase incorporates them into the nascent chain, the chain cannot be extended further. This results in the synthesis of DNA fragments ending in dideoxynucleotides of different sizes, which are loaded onto an electrophoresis gel to identify their size. Previously, four different reactions were used for each dideoxynucleotide, and each mixture was loaded into different wells of the electrophoresis gel. After completion of the electrophoresis, the bands were analyzed by autoradiography [13]. Currently, the automated method of DNA sequence analysis is used. It is based on dideoxynucleotides labeled with four different fluorochromes to the reaction mixtures, so all four reaction mixtures can be added to the capillary polyacrylamide gel electrophoresis. A scanning system detects separated fluorescent bands of DNA, and information is stored by computer [14,15]. Sanger is also suitable for metagenomics and excised DNA bands obtained from sequencing DGGE, TGGE, and T-RFLP gels [16].

In 1987, Applied Biosystems commercialized the first automated sequencer device, ABI 370 [17]. The current model, 3730xl DNA Analyzer, is a capillary array DNA sequencer with approximately a maximum insert size of >1 kilobase (Kb), a read length of 400–900 bases, 10^2^ reads per sample, and a total raw error rate of 0.001%. The main advantages of this method are a high quality and long read length. In Sanger’s strategies, amplicon sequencing is carried out in both directions, and a third read could be used to increase the sequence quality obtained [17,18].

Until the beginning of the 21st century, Sanger sequencing was the most important technology for DNA sequencing, and it was key to completing the reference sequence of the human genome in 2003 [19]. The Sanger method is still a standard for sequencing long continuous DNA (longer than 500 bases) for genome assemblies [16]. However, it is labor-intensive, time-consuming, and costly, has low throughput, and is subject to PCR and cloning bias. The Human Genome Project stimulated the development of next-generation sequencing technologies. These novel technologies replaced Sanger sequencing due to their cost- and time-effectiveness, ability to perform massive parallel analysis, and high throughput [17,20].

Despite these advances, Sanger sequencing is still considered the gold standard for species identification, especially for generating complete 16S gene sequences with the highest accuracy. Sanger sequencing use is limited to bacterial isolates, and its application in bacterial biobanks is hindered by its relatively high cost per sample. Because of that, long-read sequencing alternatives have been proposed for large-scale bacterial biobank projects [21].

## 4. A Revolution in Microbiome Analysis: The Advent of Next-Generation Sequencing

In the first 2001 draft, the estimated cost of sequencing a complete human genome was USD 100 million [22]. This financial outlay was unaffordable for independent laboratories, highlighting the need to develop mass sequencing methods. The first next-generation sequencing (NGS), also known as second-generation sequencing, was launched in the mid-2000s, claiming a 50,000-fold drop in the cost of generating a finished human genome sequence over the Human Genome Project [23]. NGS was developed and implemented during the first two decades of the 21st century. NGS is delivered by high-throughput sequencing technologies that allow millions of DNA fragments to be sequenced simultaneously, producing accurate sequence data at a reduced cost and greater speed. The main limitations of NGS approaches are that they generate a huge amount of data that must be carefully reviewed since error rates are higher (~0.1–15%) and the read length is shorter (35–700 bp) than with Sanger sequencing [17].

Although each NGS instrument has its own characteristics, they share differences from Sanger sequencing. They can work directly with DNA amplicons or total community-extracted DNA [24]. NGS is useful in taxonomic classification, unknown bacteria classification, and functional profiling of microbial communities. These methods allow a qualitative and quantitative analysis of the microbiome of different populations in different scenarios, such as in health and disease, against dietary interventions or environmental factors, among others [16,25].

The advent of NGS has brought about a real revolution in studying the human microbiome, especially since the Human Microbiome Project (HMP) was launched by the National Institutes of Health (NIH) in 2007 [26]. The HMP is an initiative funded by NIH for Biomedical Research, which aims to demonstrate that interventions can improve human health in the microbiome. The goals of the HMP are to employ new technologies for the analysis of the human microbiome in different regions of the body, to explore whether associations exist between health and disease states, to provide standardized microbiome analysis data and new methodological approaches, and to address the ethical, legal, and social implications of these studies [27].

Various NGS technologies have been developed and can be found in microbiome studies, such as 454 pyrosequencing (Roche, Basel, Switzerland), Ion Torrent (Thermo Fisher Scientific, Waltham, MA, USA), Illumina/Solexa (Illumina, San Diego, CA, USA), and SOLiD (Applied Biosystems, Foster City, CA, USA) (Figure 1a). Technologies differ in maximum insert size, read length, the scale of reads per sample, and raw error rate [28].

Sequencing by synthesis (SBS) and sequencing by ligation (SBL) approaches are available for short-read sequencing. On the one hand, SBS approaches depend on DNA polymerase and the emission of a signal (fluorophore or ionic concentration change), indicating that a nucleotide has been incorporated into the nascent DNA strand. They are usually classified into cyclic reversible termination (CRT) and single-nucleotide addition (SNA). On the other hand, SBL approaches use a DNA ligase that replaces the DNA polymerase [29,30] (Figure 1b).

SBL and SBS approaches are based on the clonal amplification of single DNA templates immobilized to a solid surface or support. Millions of individual reaction centers are used, each with a unique clonal DNA template. The information generated in the reaction centers is simultaneously collected by the sequencing platforms, allowing millions of DNA molecules to be sequenced in parallel. Having thousands of DNA copies in a limited area allows us to distinguish the emitted signal from the background noise [29,30].

### 4.1. Sequencing by Synthesis: Single-Nucleotide Addition (454 and Ion Torrent)

SNA sequencing is an SBS category that relies on a single signal to identify that a nucleotide has been incorporated into an elongating DNA strand [29]. Examples of sequencing technologies using SNA approaches include 454 pyrosequencing technology and Ion Torrent.

#### 4.1.1. 454 Pyrosequencing

Pyrosequencing technology was first published in 1993 [31]. In 2005, Rothberg and his group, through their company 454 Life Sciences, commercialized the first NGS device, the 454 pyrosequencing device [32]. Roche subsequently acquired the company. In 2013, Roche abandoned 454 pyrosequencing and acquired Genia technology, a platform based on single-molecule sequencing using nanopores [25].

Specifically, 454 pyrosequencing is sequencing by synthesis (SBS) using emulsion PCR. DNA is fragmented, and different adapters are attached to the ends. One adapter attaches the DNA to the beads for clonal amplification, and the other for sequencing. The double-stranded fragments are separated and left as single-stranded DNA. Aqueous droplets (micelle) loaded with one bead covered with complementary adapters, dNTPs, and polymerase are available. Specific conditions are set to favor that only one sequence of DNA is captured by each aqueous droplet (micelle). Emulsion PCR is performed within this micelle, and a bead coated with up to one million clonal DNA fragments is obtained [29,32].

For parallel sequencing, the beads with clonal DNA are loaded onto a PicoTiterPlate (Roche Diagnostics, Rotkreuz, Switzerland). This PicoTiterPlate contains thousands of wells; each bead remains in one well. In addition, beads that contain an enzyme mix are added to each well. Each of the four nucleotides is added individually and cyclically to each well. As a nucleotide is added to each strand, a series of reactions produces a light signal. The process begins when DNA polymerase incorporates a nucleotide into the elongating DNA strand, releasing an inorganic pyrophosphate (PPi). ATP sulfurylase uses PPi to transform adenosine 5′ phosphosulfate (APS) into ATP. Finally, luciferase uses ATP to convert luciferin to oxyluciferin and a chemiluminescent signal, ensuring that only one nucleotide is responsible for the light signal. The signal intensity depends on the number of nucleotides incorporated into regions with several identical nucleotides in a row (homopolymers). The signal intensity produced in each well is read by a charge-coupled device (CCD), and a pyrogram is generated. The process is performed in parallel and individually for each well. Analysis of the pyrogram allows the order of the nucleotides in the sequence to be determined [29,32].

Recently, 454 pyrosequencing kits progressed from single-direction sequencing with read lengths of 100 bases to paired-end with a read length of more than 250 bases and a maximum insert size of 800 bases (FLX platform) and then to paired-end with a read length of more than 500 bases and a maximum insert size of 1200 bases (FLX Titanium platform). The paired-end sequencing performed by some sequencing technologies refers to sequencing from both ends of the amplicon to increase the quality of the sequence obtained [18]. For FLX and FLX Titanium platforms, a scale of reads per sample of 10^3^ and a total raw error rate of 1% have been estimated [18,28]. The sequencing of homopolymer regions has limited accuracy because too much light is generated, which saturates the reader [29].

Although 454 pyrosequencing has been discontinued, it played a key role in developing the HMP. The HMP was intended to guide future microbiome studies, so selecting the sequencing platform was essential to maximize accuracy and consistency in 16S sequencing and profiling. Different 16S protocols were evaluated to ensure consistency in high-throughput production. Finally, the selected platform was 454-FLX Titanium, which could deliver long reads, leading to a higher taxonomic resolution. To obtain a complementary image of the taxonomic profiles, the amplified and sequenced regions were V1–V3 and V3–V5 (each longer than 500 bp) [26,33]. The 454 method was also used for other applications, such as metagenomics, to identify viral pathogens in Spanish honeybees [34].

#### 4.1.2. Ion Torrent

The Ion Torrent System was the first NGS device without optical sensing and shares the technical principles of 454 pyrosequencing technology [35]. It was developed in 2010 by Rothberg and his team in their company, Ion Torrentand later acquired by Thermo Fisher Scientific [36].

It is a sequencing-by-synthesis method that uses emulsion PCR for clonal amplification of DNA. Parallel sequencing is performed on microtiter plates with wells where each bead with clonal fragments occupies one well. Nucleotides are added one by one and cyclically to each well, and DNA elongation occurs. In this case, the difference from 454 pyrosequencing is that when a nucleotide is added to the elongating DNA strand, a proton (H^+^) is released. This release causes pH changes that are detected by an integrated complementary metal-oxide semiconductor (CMOS) and an ion-sensitive field-effect transistor (ISFET) device [37].

Ion Torrent was a paradigm shift from optical to pH variation detection, making the costs cheaper. This technology generally performs single-direction sequencing with 200 or 400 base-pair read lengths and a maximum insert size of 400 bases. A total raw error rate of 1% and the number of reads per run have been estimated depending on the platform chosen. Its limitations are shared with 454 sequencing, such as the limited accuracy in sequencing homopolymer regions [28,29].

There is a history of technological changes in Ion Torrent’s sequencing platforms, including the Ion PGM, Ion Proton, Ion GeneStudio S5, and Genexus System. Ion PGM was one of the first platforms approved for clinical use and intended for gene panels. It was followed by Ion Proton, which offered a higher throughput and extended applications to exomes and transcriptomes. Ion PGM and Ion Proton have been discontinued. Currently, the Ion PGM Dx, an in vitro diagnostic NGS platform based on the Ion PGM, is available. Subsequently, different models of the Ion GeneStudio System (S5, S5 Plus, and S5 Prime), a scalable targeted NGS offering a wide range of applications and throughput capabilities, were launched. The Genexus system has recently been launched and is the first NGS solution to incorporate an automated sample-to-report workflow that allows results reports to be generated in a single day (two user touchpoints), presenting the potential for clinical application [38].

The Ion Torrent sequencing platform is used to study microbiomes less frequently than Illumina. However, the Ion PGM and Illumina MiSeq technologies have been compared for their performance in sequencing amplicons for microbiome analysis using various sample types, 16S gene hypervariable regions, and pipelines [39]. Pylro et al. demonstrated that the same biological conclusion was obtained by sequencing the V4 region using both Ion PGM and Illumina MiSeq, employing a stringent quality filter and accurate clustering algorithms [40]. Similarly, Onywera et al. concluded that the cervical microbiome profiles obtained from Ion PGM (V4 region) and MiSeq (V3–V4 region) were generally comparable [41]. These findings were confirmed by sequencing the V1–V2 region with both platforms from a simulated community of 20 species and in human-derived samples [42]. Loman et al. concluded that MiSeq generated longer reads and lower error rates, while Ion PGM had faster response times [43].

Finally, Ion PGM has been used to analyze the microbiome in infant fecal samples by sequencing different 16S gene regions, such as V2, V3, V4, and V6, as well as combinations of these, including V3–V4, among others [44]. It has also been utilized to analyze meconium microbiome samples from neonates, examining their relationship to weight for gestational age and head circumference catch-up through sequencing the V4 region [45].

More information is available at https://www.thermofisher.com/es/es/home/brands/ion-torrent.html; accessed on 12 April 2025.

### 4.2. Sequencing by Synthesis: Cyclic Reversible Termination (Illumina)

CRT is a type of SBS category based on reversible terminator nucleotides added in a cyclic form [30]. The Illumina sequencing platforms are based on the CRT method. Illumina launched its sequencing platform in 2006 and acquired Solexa in 2007. Illumina has a wide range of sequencing instruments, from benchtop devices with low throughput to large units with ultra-high throughput, which are widely adopted by the scientific community [28].

In the Illumina system, the sample is prepared by adding different adapters to both ends of template DNA fragments. Each adapter contains different regions used for amplification, indexing (barcoding), and sequencing. The microfluid flow cell is a glass slide with channels coated with two types of oligonucleotides that are complementary to the amplification regions of the adapters. During cluster generation, the adapter region of the single DNA fragment hybridizes with one of the oligonucleotides. A polymerase then creates a complementary strand for the hybridized fragment. The resulting double-stranded molecule is then denatured, and the template strand is washed away. The DNA fragment then folds over, hybridizes with the second type of oligonucleotide, and is clonally amplified by bridge amplification. When bridge amplification finishes, reverse strands are cleaved and washed off, leaving only forward strands, resulting in clonal amplification of all DNA fragments [29,30].

Sequencing begins by extending a primer hybridized to the sequencing adapter region by adding modified nucleotides. Each modified nucleotide has a 3′-terminating group (dNTPs) and is labeled with a different cleavable fluorophore. The four modified nucleotides are added, and as they are blocked at 3′, only one is incorporated. Then, the unincorporated dNTPs are removed. Images are captured and analyzed to identify which dNTP has been incorporated. Then, the terminating group and the fluorescent dye are cleaved. Following that, a new cycle can start [29,30].

Illumina technology has several available platforms, such as iSeq 100, MiniSeq, MiSeq, NextSeq Series, and NovaSeq Series, which are commonly used in microbiome studies. Illumina platforms usually generate paired-end reads with 250 bases per readand around 50,000–100,000 reads per sample [46]. Estimated error rates range from 0.1% to less than 1% and the reads per run depending on the platform chosen. Illumina platforms exhibit some bias in AT- and GC-rich regions and a propensity for substitution errors [29]. Equipment is expensive and requires a high DNA concentration [28].

Illumina platforms have been widely used in microbiome studies. For example, the Illumina GAIIx platform with 101 base paired-end reads was used to asses microbiome function in the HMP [26,33].

For more information, consult the following website: https://www.illumina.com/; accessed on 12 April 2025.

### 4.3. Sequencing by Ligation (SOLiD)

SOLiD is a sequencing method that does not use DNA polymerase to create the complementary DNA strand. This method is based on the use of DNA ligase and fluorescently labeled probes. A primer is attached to the DNA template. The labeled probe binds to its complementary sequence adjacent to the DNA-primed template [30].

Fluorescently labeled probes are known as 8-mer probes. These are probes in which the first nucleotide (one-base-encoded probes) or the first and second nucleotides (two-base-encoded probes) are designed with the possible combinations of the elongating DNA strand. The third to fifth nucleotide bases are degenerate, and the sixth to eighth are universal, allowing interaction with different template sequences. The eighth nucleotide is fluorescently labeled. These probes identify the first nucleotide or the first two nucleotides adjacent to the hybridized primer [30].

The SOLiD sequencing platforms use the sequencing-by-ligation method. Applied Biosystems introduced the SOLiD (Sequencing Oligonucleotide Ligation and Detection) platform in 2007 through Life Technologies. It is based on ligase enzymology, primer reset functionality, and two-base-encoded probes. It is a ligation sequencing method that uses emulsion PCR for clonal amplification of DNA [28].

Sample preparation is similar to 454 sequencing. DNA is fragmented and denatured. Adapters are attached to the ends. One adapter is used for DNA binding to the beads for clonal amplification, and the other for sequencing. Emulsion PCR is performed, and clones of the template DNA are obtained by coating a magnetic bead. The beads are loaded with clonal DNA on a glass slide for parallel sequencing. Next, a universal primer complementary to the adapter sequence of the DNA template is added, and the primer hybridizes with the DNA template, generating a site at which to initiate ligation with labeled probes [29]. A mix of two-base-encoded probes is then added. If the first two nucleotides of the probe are complementary to the DNA template, the probe hybridizes with the DNA template. DNA ligase is added to join the probe to the primer. Free probes are washed away, and a laser detects the fluorescent signal to identify the first two nucleotides. The ligated probes are cleavable after the fifth nucleotide with silver ions to remove the fluorescent dye and regenerate the 5′-PO4 group for subsequent ligation cycles [47]. This cycle is repeated ten times, resulting in ten color calls spaced at five-base intervals. A fraction of the DNA sequence is generated because nucleotides three through five are unknown for each five-base group. The primer is stripped from the DNA template, and a new ligation round is carried out with primers of length n-1 to ascertain the complete sequence of the template DNA molecule. Three more rounds of ligation cycles with n-2, n-3, and n-4 primer lengths are generated [47]. Color calls of five ligation rounds are ordered and analyzed to decode the template DNA sequence [30].

SOLiD platforms perform single-end reads with around 50–75 base lengths. A total error of less than 0.1% has been estimated, and the amount of reads per run depends on the platform chosen. SOLiD platforms have some bias in palindromic regions and are relatively slow, and the read length is shorter than other methods, which limits their wider applications [28].

SOLiD was first feasibly and cost-effectively employed in 2013 to perform 16S gene and shotgun sequencing of human gut microbiome samples [48]. However, it is rarely used in microbiome research and has been replaced by other sequencing technologies.

If more information is required, consult the following website: https://www.thermofisher.com/es/es/home/brands/applied-biosystems.html; accessed on 12 April 2025.

## 5. New Era in Microbiome Analysis: The Development of Third-Generation Sequencing

It has been described that genomes have long repeat-rich elements and copy number variation with an important role in human diseases, evolution, and genetic diversity [49,50]. Many of these elements are so long that second-generation sequencing technologies, based on short-read sequencing, cannot identify them. On the other hand, bacterial taxonomic classification accuracy is dependent on amplicon length [51] and on single-nucleotide polymorphisms (SNPs) [52].

The 16S gene is ~1500 bp in length and contains nine variable regions. The complete sequencing of this gene allows a higher taxonomic resolution of microbial communities, reaching the species and even strain level [53], as well as the identification of potential SNPs of bacterial strains in association with clinical relevance [52]. However, second-generation sequencing technologies do not allow for the analysis of the entire 16S gene [53].

The search for a better balance between throughput, read length, and cost of analysis has led to the development of third-generation or long-read sequencing technologies. These technologies perform single-molecule, real-time sequencing (Figure 1a). They use long individual DNA molecules for sequencing, without the need for a clonal amplification step, which results in a reduction in biases associated with the amplification process and the ability to obtain long and continuous DNA reads (more than 10 kb) (Figure 1b). The main limitation of these technologies is that they have a higher error rate than short-read sequencing technologies, which can be compensated for with a greater sequencing depth. The difference in the error rate between the long- and short-read sequencing technologies is decreasing, with them becoming highly comparable in some cases [28].

Long-read sequencing has led to an improved resolution in genomic research, notably by enabling the analysis of the entire 16S gene [53] and the analysis of regions with large repeats and copy number variations [29].

Indeed, as noted in an article published by Marx in the *Nature Methods* journal, long-read sequencing technologies have been selected as the 2022 Method of the Year [54]. The Vertebrates Genomes Project [55], the Telomere-to-Telomere Consortium (T2T) [56], and the Human Pangenome Reference Consortium (HPRC) [57] are using long-read sequencing.

### 5.1. Pacific Biosciences

In 2010, Pacific Biosciences (PacBio) developed the first third-generation sequencing method, which uses a single-molecule real-time sequencing (SMRT) approach [17].

It is a sequencing approach using an immobilized DNA polymerase, fluorescent molecules as in NGS, and real-time analysis of the signals generated by incorporating nucleotides into the elongating strand. SMRT has become one of the third-generation sequencing platforms most employed in NGS [28].

Sequencing is performed on a chip (SMRT cell) with embedded zero-mode waveguide (ZMD) nanostructural arrays. The ZMD is a cavity with attached polymerase at its bottom, and a single-stranded DNA template molecule is advanced through the ZMD. SMRT cells are added to the four different fluorescently labeled nucleotides. When one of the nucleotides is incorporated into the growing strand, the real-time camera generates and records a light signal. While incorporating the nucleotide into the complementary strand, the polymerase can release the fluorophore from the previously incorporated nucleotide [58].

With PacBio, both paired and single ends can be achieved. The main advantages of PacBio over second-generation sequencing technologies are rapid sample preparation, no need for a PCR pre-step (which reduces amplification bias), a fast read rate, and a read length of tens of kilobases [17,59]. PacBio has low throughput and low flow cell success and is less cost-effective than other sequencing platforms [59].

PacBio has developed HiFi (high fidelity) sequencing, which generates a consensus sequence through multiple passes of a single circular template molecule to improve the accuracy of SMRT sequencing. It generates long HiFi reads with an average length of 13.5 kilobases (kb), reaching 99.9% accuracy and higher [60]. HiFi genomic applications are haplotype phasing, variant detection, genome assembly, and epigenetics. Indeed, HiFi technology is a very good option for large-scale research, such as the HPRC project funded by the National Human Genome Research Institute (NHGR). For laboratories, both cost and accuracy are essential. The main limitation is that this technology is not affordable for most labs. Both long-read sequencing platforms, Oxford Nanopore Technologies (ONT) and HiFi, were co-awarded as methods of the year in 2022 by the journal *Nature Methods* [54].

PacBio offers long- and short-sequencing platforms. Vega and Revio are long-read sequencing systems that generate HiFi reads. The Onso system, based on SBB chemistry, is the only short-read sequencer that produces Q40+ data [61]. The PacBio platform was used to assess the origin of the Haitian cholera outbreak [62] and to examine the oral microbiome of healthy Chinese children [63].

Additional information can be found at https://www.pacb.com/technology/sequencing-by-binding/; accessed on 29 April 2025.

### 5.2. Oxford Nanopore Technologies

In 2014, the prototype nanopore sequencer MinION from Oxford Nanopore Technologies (ONT) came to the market. Unlike other platforms, nanopore sequencers are not based on detecting the signal generated by incorporating nucleotides into an elongating DNA chain. In this case, single-stranded DNA molecules are directly sequenced by passing through a nanopore. An amplification step is not always required for library preparation [28].

ONT sequencing devices employ flow cells containing an array of protein nanopores embedded in an electro-resistant artificial membrane where a voltage is applied. Each nanopore is connected to a channel and a sensor chip that measures the electrical current passing through the nanopore. A secondary motor protein is associated with the nanopore and assists a single DNA molecule in passing through the nanopore. As the DNA molecule passes through the nanopore, there is a change in the electrical current. This change in electrical current is then decoded using basecalling algorithms to establish the DNA or RNA sequence in real time [64].

ONT devices generate short to ultra-long (>4 Megabases (Mb)) reads [65]. MinION mk1B is the only pocket-sized portable sequencing device with 512 nanopores and can be connected directly by USB 3.0 to a computer for data collection [29]. ONT has also released high-throughput platforms: GridION, which operates with five MinIONs, and PromethION, which works with 24 or 48 flow cells, each with 2675 nanopores. These platforms allow very long genomes (>100 kb) to be sequenced in a cost-efficient manner [66].

Nanopore sequencing can achieve 1D, 2D, and 1D^2^ reads. One-dimensional sequencing employs nanopores where only one strand of the DNA molecule is sequenced, while the other is discarded. Two-dimensional sequencing is based on using a hairpin structure at the end of the double-stranded DNA to join the template and the complementary strand. Thus, the template strand, followed by the hairpin and the complementary strand, is sequenced sequentially through the nanopore, equivalent to sequencing a DNA molecule twice. In 1D^2^ sequencing, the template strand and the complement are also sequenced, but they are linked with a special adapter instead of a hairpin [67].

ONT allows long read lengths, portability, and real-time analysis [28]. The cost per base is cheaper than PacBio. The sequencing accuracy of ONT’s platform is close to 99% for simplex reads at a base level. With the upgrades in chemistry, nanopores of the R10.4.1 flow cell, and the improved basecaller, an accuracy over 99.9% for duplex reads has been achieved [68,69]. ONT is applied in genomics for phasing, assembly, structural variant detection, single-nucleotide and insertions and deletions (indel) analysis, and methylation. As we have mentioned, both long-read sequencing platforms, Oxford Nanopore Technologies (ONT) and HiFi, were co-awarded as the methods of the year 2022 by the journal *Nature Methods* [54]. For laboratories, not only accuracy but also cost is essential. Steven Salzberg, a researcher at Johns Hopkins University, has considered ONT “with few exceptions, the only one real choice for long-read sequencing” due to the high accuracy, the cheaper cost per base, and the availability of portable devices [54].

ONT technology has been used for sequencing the *Bacillus velezensis* TS5 genome from Tibetan sheep feces and studying its potential as a probiotic [70] and also for sequencing the intestinal microbiome of neonates, with promising results for detecting pathogens in neonatal clinical settings [71].

Additional information can be found at https://nanoporetech.com/; accessed on 29 April 2025.

## 6. Long-Read Sequencing Technologies: New Perspectives in the Analysis of Microbiome

Long-read sequencing technologies such as PacBio and ONT have provided novel and complete information on previously only partially characterized complex microbial communities.

These sequencing platforms are flexible to the sample type and can sequence microbial DNA obtained from feces, saliva, environmental, or other samples. They offer alternatives for adapting to difficulties during research, such as sequencing bacterial DNA from samples containing a high proportion of human DNA, as in colorectal cancer tissue [72,73].

Long-read sequencing technologies enable an accurate taxonomic resolution down to species and even strain levels of the microbial community, overcoming a limitation for short sequencing technologies. Long-read platforms allow complete sequencing of the prokaryote 16S gene and even fungi ribosomal operons (16S-ITS-28S) [74,75]. Moreover, long-read sequencing platforms have been a breakthrough in virome research, allowing the analysis of complete viral genomes [76,77,78].

Long-read sequencing platforms have also expanded the possibilities for retrieving whole genomes and functional inference of the microbiome in diverse ecosystems by recovering high-quality metagenome-assembled genomes (MAGs) [79].

These technologies offer an additional dimension to the conventional taxonomic and functional analyses of microbial communities, as they allow the direct detection of epigenetic modifications during the sequencing process [80,81].

### 6.1. Predominance of Host DNA: A Challenge in Microbiome Analysis

Samples with high host-to-microbial DNA ratios represent a significant challenge in microbiome analysis, reducing the sequencing coverage of microbial genomes and complicating subsequent taxonomic and functional analysis [72,73].

When sequencing stool samples, the microbial community is expected to be accurately represented, as host DNA represents a very low amount (<10%). However, in samples such as saliva, throat, buccal mucosa, and vaginal swabs, where host DNA exceeds 90%, detection of low-abundance species is compromised [33,73,82]. This proportion is exacerbated in tissue samples, where human DNA makes up 97–99% of the readings, making it difficult to detect both low- and high-abundance microbial species [83].

In these cases, sequencing the entire genetic material of the sample can be inefficient and costly. Targeted sequencing strategies have therefore been developed to reduce the time spent sequencing regions that are not of interest, the sequencing costs, and the data generated. There are different strategies to carry out targeted sequencing, such as amplicon sequencing and adaptive sampling.

In this sense, long-read lengths can increase the number of specific primer binding sites, which is limited in short-read lengths [6]. In addition, ONT has developed an innovative software-based targeted sequencing strategy, adaptive sampling. This method is based on ONT’s real-time sequencing. As the sequence passes through the nanopore, the system identifies whether it contains a region of interest and selects target sequences. This method is integrated into MinKNOW, the operating software that controls all ONT sequencing platforms. Adaptive sampling can be run in two modes: enrichment or depletion. In the enrichment mode, a bed file containing the regions of interest and a FASTA file containing the reference are loaded into MinKNOW. In the depletion mode, a file is loaded with regions that are not of interest (e.g., host DNA in a metagenomic microbiome study). Adaptive sampling allows ~5–10-fold enrichment for regions of interest, improving efficiency in samples highly contaminated with host DNA [84].

### 6.2. Towards a More Accurate Taxonomic Identification in Microbial Communities

Molecular identification of species is carried out using marker genes. These genes are selected based on the ease of amplification, taxonomic resolution, and the existence of a reference database for their classification [6].

Short-read sequencing has been widely used to profile microbial communities in various ecological settings [85]. These investigations have allowed alpha and beta diversity analyses, but the taxonomic and functional resolution has been limited due to the short length of the sequenced fragments [74,75]. However, long-read sequencing technologies yield an accurate taxonomic resolution of species and even strain levels of the microbial community [6,86]. This improvement in resolution has been especially valuable for accurately identifying bacteria, fungi, and viruses of the microbial communities.

#### 6.2.1. Bacteria

In bacteria, the most commonly used taxonomic marker for identification and phylogenetic classification is the 16S gene. The introduction of 16S full-amplicon sequencing in microbial ecology research has become a powerful approach providing high-resolution bacterial taxonomy [87]. For example, whole 16S gene sequencing with PacBio allowed the identification of stronger gut microbiome associations in obese children with the risk of steatotic liver disease compared to sequencing of the V3–V4 regions [88].

#### 6.2.2. Fungi

In fungi, the most commonly used taxonomic marker is the ITS region of the rRNA. However, other rRNA regions, such as 18S and 28S rRNA, are used for the taxonomic classification of various fungal phyla [89]. The ITS region is typically between 500 and 700 bp, and most studies using short-read sequencing analyze the ITS1 or ITS2 region, which ranges between 250 and 400 bases and has a large variation between groups. In many fungal taxa (e.g., in the order Hypocreales), only one ITS region has sufficient variability to identify species [6]. Analysis of the ITS2 sub-region usually results in less taxonomic bias than ITS1 because it has less length variation and more universal primer binding sites [90].

Long-read sequencing platforms allow the analysis of the entire ITS region and part or all of the flanking rRNA genes, such as 18S or 28S. Universal primers have been designed to allow amplification of the fungal ribosomal operon in one 10 kb amplicon or two 5 kb amplicons. This strategy allows the inclusion of all ribosomal markers: the external transcribed spacer (ETS), small subunit (18S), ITS1, 5.8S, ITS2, large subunit (28S), and intergenic spacer (IGS) [75,91].

Analyzing the ITS region rather than sub-regions has important advantages, as it results in a higher taxonomic resolution and less amplification of dead organisms. However, this approach performs poorly on low-quality samples, such as herbarium specimens, where the DNA has degraded, and it is not easy to preserve full-length ITS regions [89,92].

The development of long-read sequencing platforms has made it possible to delve deeper into the richness and composition of fungal communities and specific fungal groups. Sequencing of the full operon of the rRNA gene with PacBio has expanded new fungal taxa at the order level by 10–20% by phylogenetically locating taxa that had not previously been identified with ITS regions alone [93]. Furthermore, ONT sequencing of complete ribosomal operons allowed rapid, real-time identification of fungi down to the species level in otitis externa samples from dogs [94].

#### 6.2.3. Virome

Advances in NGS technologies and specialized bioinformatics tools have enabled further research into the human virome. Although the gut microbiome is mainly composed of bacteria and archaea (more than 99% of the biomass), fungi, protozoa, and viruses are also present. It has been suggested that commensal viruses, such as phages and DNA and RNA viruses, are found in the healthy human gut [76]. It has been estimated that there are 10^9^–10^12^ individual virus particles per gram of human feces [95,96].

The virome (phages and other host viruses) exerts an important intestinal physiology and immune system functions. In this context, long-read sequencing technologies are advancing the investigation of virome composition and functions by capturing almost complete viral genomes and even detecting epigenetic modifications directly in viral genomes [76,77,78].

However, virome research presents several challenges. The first protocols to optimize the extraction of viral DNA from human feces and the subsequent workflow are being developed to enable long-read sequencing and improve virome resolution [76,77,78].

### 6.3. Complete and Accurate Assembly of Microbial Genomes

Short-read sequencing has long been the preferred method for generating reference genomes, especially in pure culture and metagenomic studies. However, this technology has limitations in the resolution of repeat regions that exceed the insert size of the libraries. This limitation is exacerbated in metagenome samples, where phylogenetically related species may have long, nearly identical DNA sequences [97]. As a result, short-read studies often fail to sequence amplicons larger than 300 bp, perform complete assemblies, or abandon analyses due to data fragmentation. In the face of these challenges, long-read sequencing has become more popular for analyzing pure cultures and metagenomes [79,98].

Metagenomics studies using long-read sequencing technologies such as PacBio and ONT have revolutionized the reconstruction of MAGs by generating reads longer than 10 kb. These long reads improve the contiguity of assemblies, which is essential for understanding the structure and function of microbial genomes. This improvement in assembly is especially visible in genomic regions that are difficult to assemble with short-read sequencing, such as repetitive sequences, mobile genetic elements, and regions with extreme GC contents or structural variations [98,99].

In particular, PacBio HiFi reads, which offer low error rates and relatively long read lengths, can generate near-complete microbial genomes [100]. However, their high cost per base represents an economic challenge for many researchers [97]. In contrast, long-read sequencing with ONT has democratized the sequencing of microbial genomes, making it easier to obtain highly contiguous genomes from pure cultures or metagenomes. However, to achieve near-complete genomes, it has traditionally been necessary to use short-read polishing to correct indels in homopolymeric regions [101,102].

Nonetheless, a recent study has shown that the ONT 10.4.1 technology can generate near-complete microbial genomes from isolates or metagenomes at 40× coverage without polishing short reads or a reference genome. The MAGs generated were in the same IDEEL score range as those from PacBio HiFi. Although long homopolymers (≥10 bases) will continue to be a challenge, they are only a small fraction of microbial genomes [97].

With advances in accuracy (now reaching up to 99.9% with PacBio HiFi and 99% with ONT R10.4.1) and cost reductions, long-read sequencing is expected to become more widely used in microbiome studies, allowing complex genomes to be resolved with greater accuracy and depth [99].

An illustrative example is the use of ONT sequencing to assemble a high-quality genome of a *Mycoplasma* species from the human intestine, which could not be properly assembled with short reads due to its low GC content. This more detailed approach to structural variants allows the investigation of their effect on microbial communities [103]. Long reads have also been used to assemble the initial reference genomes of non-model organisms, such as *Rhizoctonia solani*, a plant pathogenic fungus species, from ONT reads [104]. Another use of long reads is to close genomic gaps in species with reference genomes. For example, 217 high-quality complete genomes of *Salmonella enterica* have recently been generated from PacBio long reads, contributing to the expansion of genomic resources for surveillance [105]. In some complex cases, long reads have been combined with short reads to assemble initial reference genomes of species or close gaps in reference genomes [106,107].

### 6.4. Microbial Epigenome Profiling

Detecting epigenetic modifications directly on microbial genomes is another application of long-read sequencing technologies that cannot be achieved with short-read platforms. Bacterial epigenetics, particularly DNA methylation, is essential in adapting bacteria to their environment [108,109]. While eukaryotic epigenetics has been investigated, bacterial epigenetics remains largely unknown. In contrast to eukaryotic epigenetics research, which has focused on the study of 5-methylcytosine (5 mC), bacteria also have other important modifications, such as N6-methyladenine (6 mA) and N4-methylcytosine (4 mC) [110,111,112].

Bacterial DNA methylation was discovered by studying the regulation of restriction–modification (RM) systems. These systems comprise endonuclease and methyltransferase enzymes with common target DNA sequences. The endonuclease can cut the DNA sequence if the methyltransferase has not methylated it. It has been proposed that these RMs could protect bacteria from exogenous DNA sequences [112].

In addition, other processes regulated by bacterial DNA methylation have been described, such as pathogenicity, DNA repair, chromosome replication and segregation, cell cycle control, and even reversible switching of gene expression [113,114,115,116,117]. Due to the importance of epigenetics in the functions and dynamics of microbacterial communities, research on bacterial epigenetics is an emerging field for microbiota modulation [109].

Traditionally, a method combining methyl-sensitive restriction enzyme digestion and NGS has been used to profile bacterial epigenomes [118]. Although this method provides information on methylation patterns at specific loci, it is limited by the specificity of the enzymes. Therefore, complementary techniques such as whole-genome bisulfite sequencing (WGBS) [119] or immunoprecipitation sequencing of methylated DNA (MeDIP-seq) are often required [120].

Long-read sequencing techniques have facilitated the study of the epigenome by allowing direct detection of DNA methylation patterns. While PacBio generates a detailed profile of bacterial epigenomes using unique fluorescent signals during DNA synthesis, ONT relies on ionic current interruptions as the DNA sequence passes through the nanopore, which requires fewer resources [80,81,121]. One of the main differences between the two technologies is that PacBio can detect modifications such as 6 mA and 4 mC, but not 5 mC, while ONT can detect all of them. On the other hand, PacBio has a high accuracy of single-read methylation but is technically more complex and expensive. With ONT, bioinformatics tools are needed to optimize accuracy, but it offers a portable and more affordable option [122,123]. Therefore, the choice between these platforms depends on the specific characteristics of each research study [109,124].

## 7. Desirable Characteristics for Microbiome Sequencing Methods

Among the most widely used high-throughput sequencing platforms for microbiome analysis in ecosystems worldwide are Illumina, PacBio, and ONT. Each technology offers specific advantages and limitations, so the choice of the most appropriate technology will depend on the requirements of each project. In this context, the following question arises: What characteristics should be considered when selecting the sequencing method that best suits the research objectives? Next, we discuss desirable features that could inform this decision.

### 7.1. Read Length

The remarkable genomic plasticity of bacteria, largely attributable to horizontal gene transfer, means that specific functions and traits are linked to genomic regions particular to each species and strain [125,126]. Because of this genomic plasticity, a main challenge nowadays is achieving the most accurate bacterial taxonomic classification possible. Different species of the same genus can be functionally divergent, so this accuracy is especially important when characterizing the microbiome in certain phenotypes or clinical studies. Therefore, achieving taxonomic resolution at the species or even strain level is essential for functional inference and accurate ecological trait assignment [74,127].

The two most common microbiome profiling methods are amplicon and metagenomic shotgun sequencing. So far, amplicon sequencing, which focuses on a specific gene or region of the genome (such as the 16S gene for prokaryotes), remains the most widespread and cost-effective strategy. Its advantages include high sensitivity, a reduced risk of host contamination, the ability to identify and reduce false positives, access to data analysis platforms (such as QIIME 2 and EPI2ME), and a lower cost than shotgun sequencing [74,128,129].

Numerous studies have debated which region of the 16S gene provides the most accurate taxonomic resolution. It is currently considered that sequencing the entire 16S gene (~1500 bp) yields more accurate results than choosing specific regions [6,86]. Using long-read sequencing methods has made identifying species or strain levels possible [53]. However, short-read sequencing platforms like Illumina still dominate the market. As these platforms cannot cover the complete 16S gene sequences, most studies employ fragment amplification to a length of 100–550 bp. In this context, amplicons up to 300 bp can be fully sequenced with paired 2 × 300 bp reads. However, a minimum overlap of ~50 bp is needed for longer amplicons to ensure reliable assembly. This length limitation generally reduces the taxonomic resolution to the genus level or higher [130].

PacBio and ONT sequencing platforms generate entire 16S gene reads (~1500 bp) and have the potential to provide a high taxonomic resolution and accuracy at the species and strain levels of bacterial communities. In comparison, Illumina platforms can generate sequencing reads of ≤ 300 bp, which allows the analysis of different regions of the 16S gene limited to genus or higher taxonomic levels [131] (Table 1).

Moreover, regarding the MAGs, PacBio and ONT can generate reads longer than 10 kb, revolutionizing the reconstruction of complete and accurate MAGs [98,99]. However, fragment lengths are limited to around 100–550 pb in Illumina platforms [97]. In a comparative metagenome assembly experiment, the same fecal reference sample (ZymoBIOMICS) was sequenced with ONT and Illumina, reaching paired read depths (~100 Gb). ONT recovered ~2.5 times more high-quality MAGs than Illumina. Regardless of the sequencing depth, Illumina recovered no contigs >1 Mb and no closed MAGs, while ONT generated 935 contigs >1 Mb and 58 closed MAGs [132]. Also, when comparing ONT to PacBio in the same fecal sample, ONT generated ~1.8 times more high-quality MAGs per flow cell and ~1.5 times more closed MAGs. For the equivalent recovery of high-quality MAGs, 25% of the PromethION flow cell was required, but 100% of Revio [132].

It is essential to note that the limitation of a resolution down to the genus level of short-read methods cannot only be attributed to the limitations of the sequencing platform but also to the characteristics of the database used, as well as the taxonomic resolution of the 16S gene itself.

On the one hand, from databases such as SILVA, RDP, Greengenes, or NCBI, it is possible to assign sequences to Operational Taxonomic Units (OTUs) or Amplicon Sequence Variants (ASVs). The taxonomic resolution achieved varies depending on the database size (number of taxa) and the resolution capacity (classification level) [133]. In the case of long-read sequencing, a high-resolution database is needed. For example, if 16S gene long-read sequencing results are classified using the SILVA database, which primarily focuses on covering short-read sequences, higher-resolution information may not be fully captured, and in some cases, around 30% of the reads could not be correctly classified [134]. Moreover, the taxonomic resolution also depends on the classification tool, and it may be higher and faster when using Kraken 2/Bracken rather than tools such as QIIME 2 [135]. Therefore, the choice of sequencing platform is important; however, the taxonomic resolution ultimately varies depending on the database and classification tool used. For this reason, depending on the choices made, even researchers studying the same topic may obtain different results.

On the other hand, in cases where the evolutionary relationship is very close, the taxonomic resolution is limited by the 16S gene itself. For example, the 16S gene sequences of the genera *Escherichia* and *Shigella* are 99.7% identical [136,137]; however, differences exceeding 3% are typically considered species-specific [138]. For example, in some cases, databases such as SILVA choose to group the genus name as *Escherichia/Shigella* or *Allorhizobium-Neorhizobium-Parhizobium-Rhizobium* [139,140]. In these situations, the 16S gene taxonomic resolution should be considered for accurate identification.

Therefore, these limitations outside the sequencing platform have to be taken into account, which highlights the complexity of the taxonomic classification process.

### 7.2. Accuracy

This section discusses the underlying sequencing errors, types of errors, strategies to achieve higher accuracy, and the current accuracy for Illumina, PacBio, and ONT sequencing platforms (Table 2).

Illumina platforms have long been recognized for having a high basecall accuracy of 99.9% for most bases. Illumina’s website describes each platform’s technical specifications, indicating the percentage of bases with a quality score (Q score) higher than 30 (Q30 corresponds to a 99.9% accuracy). The predominant source of error is library construction, followed by sequencing-related errors and DNA damage [141]. A common error pattern on these platforms is that the base immediately following a base-specific homopolymer is substituted for the base that makes up that homopolymer. A higher error rate has been observed for G/C homopolymers than for A/T homopolymers. Some instruments do not follow this pattern; e.g., in NovaSeq6000, the pattern appears reversed [142]. Different error prevention and correction strategies have been proposed to improve Illumina’s accuracy. For example, Illumina has launched the XLEAP-SBS chemistry [143], compatible with sequencing platforms such as MiSeq i100 or NextSeq 1000/2000 (2 × 300 bp) exceeding 85% of bases with Q30+ [144].

Long-read sequencing platforms have traditionally had higher error rates than short-read sequencing, so different error correction strategies have been developed [145].

In the case of PacBio, the initial errors were mainly due to incorrect interpretations of the fluorescence signals and random errors during polymerase synthesis. In the PacBio data, low error rates were reported for substitutions (1.7%) (with a predominance of A↔C and G↔T transversions), intermediate for deletions (3.2%), and high for insertions (8%) [145]. However, the new PacBio CCS (consensus circular sequencing) protocol generates HiFi reads with an accuracy of more than 99.9% [146]. This method involves repeatedly sequencing the same molecule to create a consensus, which improves accuracy and also leads to higher costs and lower overall yields [145]. Although these values may vary depending on the experimental factors involved, it has been reported that approximately 95% of bases in a 0.5–5 kb library have a Q30+ score, whereas it is 90% in a 10–15 kb library [147].

In the case of ONT, the initial errors were mainly concentrated in homopolymeric regions due to the nanopore design, which biases the recognition of the current signal with A/T bases. Therefore, error rates of 4% (±0.5%) were reported for substitutions, deletions, and insertions. The most frequent substitutions were of the A↔G and C↔T transition types [145]. However, ONT has improved the accuracy of its readings with the Kit 14 chemistry, R10.4.1 flow cell, and super-accurate basecaller. These upgrades allow duplex sequencing of both strands of the same DNA molecule, improving accuracy in homopolymeric regions and achieving a Q30+ score for duplex reads. The upgrades generate simplex reads with a Q20+ score [68,69]. In addition, some tools can improve the accuracy of ONT readings by using reference or short readings, although this introduces more complexity to the analysis [97,145].

### 7.3. Runtime

Short sequencing times are especially important in longitudinal studies and clinical or public health settings. The following section explains the manufacturer-reported runtime associated with the maximum theoretical throughput on different sequencing platforms.

For Illumina platforms, the reported runtimes range at about 15 h for MiSeq i100 (2 × 300 bp), 34–42 h for NextSeq 1000/2000 (2 × 300 bp) (depending on the flow cell type P1 or P2), and about 38 h for NovSeq 6000 (2 × 250 bp) [144]. PacBio, Vega, and Revio systems offer runtimes of about 24 h [148]. In the case of ONT, the reported runtime is 16 h for Flongle, while it is up to 72 h for MinION, GridION, and PromethION [149,150] (Figure 2).

A feature that should be mentioned about ONT flow cells is their flexibility. Although they contain sufficient buffers to run continuously for up to 72 h, the user can stop the run anytime. Therefore, the user can alternate between running continuously and stopping the sequencing when they choose (e.g., when sufficient data has been generated), performing a wash of the flow cell, and loading a new sample until the buffer and nanopores are exhausted [151].

### 7.4. Sequencing Output per Cell

This section compares the theoretical maximum output per cell per run for some sequencing platforms that could be commonly used for microbiome analysis in ecosystems. Outputs can differ depending on the type of library, the sequencing cell type, and the conditions under which the experiments are carried out (Figure 3).

For Illumina platforms, MiSeq i100 with a 25 M flow cell (2 × 300 bp) generates a theoretical max output of 15 Gb with around 15 h of runtime. The NextSeq 1000/2000 with a P1 flow cell (2 × 300 bp) generates 60 Gb with a 34 h runtime, and a P2 flow cell (2 × 300 bp) generates 180 Gb in 42 h. The NovaSeq 6000 SP flow cell (2 × 250 bp) produces 325–400 Gb for around 38 h. The maximum number of flow cells that can be placed simultaneously is 1 in MiSeq i100, 1 in NextSeq 1000/2000, and 2 in NovaSeq 6000 [144].

For PacBio, the theoretical max output for the Vega system with the SMRT Cell 8 M is 60 Gb for a 24 h runtime, while for the Revio system, the SMRT Cell 25 M has a max output of 120 Gb. At a time, the Vega can analyze a maximum of 1 SMRT Cell 8 M (60 Gb), while Revio can analyze 4 SMRT Cell 25 M (total output 480 Gb) [147,152].

In the case of ONT, the theoretical max output for the Flongle flow cell used in MinION is 2.6 gigabases (Gb) for 16 h of runtime. For the flow cell used in MinION and GridION, the theoretical maximum output over 72 h of runtime is 48 Gb, whereas for the PromethION flow cell, it is 290 Gb. The maximum number of flow cells that can be placed at a time is 1 in MinION (48 Gb), 5 in GridION (total output 240 Gb), and commonly 24 in PromethION (total output 6600 Gb) [149].

Differences in sequencing speed (Gb/hour) were observed between the various platforms analyzed, depending on the model and execution conditions. Illumina platforms offered intermediate speeds ranging from 1.0 Gb/h for MiSeq i100 to 19.08 Gb/h for NovaSeq 6000. PacBio platforms showed comparable speeds, achieving 2.5 Gb/h for Vega and 20 Gb/h for Revio. Finally, the ONT systems showed theoretical sequencing speeds ranging from 0.16 Gb/h for Flongle to 91.67 Gb/h for PromethION, the fastest sequencing platform. These results suggest that the sequencing speed depends on the platform used. Therefore, it should be considered when choosing a platform, especially for studies requiring a high sequencing depth, large genomes, numerous samples, or fast results.

### 7.5. Cost

A low cost per microbiome sample is essential in performing studies with many samples and implementing them in public health contexts. Figure 4 shows the approximate costs of sequencing instruments and cells. The prices are approximations, calculated in US dollars (USD) based on manufacturer information, and are indicated with $ in the figure. They may vary by geographic region and supplier.

Each sequencing platform requires specific reagents. Moreover, different kits or protocols may be used for the same platform depending on the research objective and the genetic material’s characteristics. This variability makes it complex to give an accurate cost per base, especially when protocols involve expensive third-party reagents. In general terms, the estimated cost per Gb is usually lower for ONT (~12–13 USD/Gb) than for PacBio (~17–100 USD/Gb), while for Illumina, it is quite variable depending on the platform and amplicon size (~9–175 USD/Gb) [54,90]. It is also important to note that the sequencing depth directly affects sensitivity. Increasing the sequencing depth comes at a high cost, and in the case of metagenomics, the most cost-effective alternative is usually Illumina [153,154]. For sequencing amplicons, ONT is often a cheaper alternative, although this also depends on the diversity of organisms and their abundance, as a greater depth of sequencing is required to detect minority taxa [155].

Figure 4, therefore, includes the price of the sequencer and the cell, an important laboratory consideration. The instrument price can be very high, which may necessitate outsourcing the sequencing [29,156]. The price of the cell is included because, regardless of the platform selected, it will always be a necessary reagent, with fewer options available, and one of the main reagents in any sequencing reaction.

### 7.6. Equipment Portability

The Illumina and PacBio sequencing platforms are bulky benchtops or floor-standing equipment closer in size to a domestic fridge-freezer [144,148]. The ONT sequencers are table-top devices, where the largest is similar in size to a microwave oven (PromethION) and the smallest fits in the palm of a hand (MinION) [149,150]. A comparison of the portability level for the different sequencing platforms is presented in Table 3.

MinION’s small size makes it a state-of-the-art sequencer for portable sequencing, especially useful outside the traditional laboratory environment. MinION only requires connection to any laptop via a USB port, making it an affordable and portable alternative to the cumbersome and time-consuming process of transporting samples to distant labs. A prime example of its usefulness was during the 2015 Ebola pandemic in West Africa. MinION was used to sequence the Ebola virus genome in real time in resource-limited settings, allowing outbreaks to be monitored quickly [157].

### 7.7. Bioinformatic Tools for Sequencing Data

Bioinformatics analysis is essential in interpreting the data due to the large amount of data generated by massive sequencing platforms. The existence of bioinformatics tools to process the data generated by each sequencing platform and the possibility of automating these processes represent an added value, especially in public health contexts where efficiency and scalability are required.

The development of bioinformatics analysis software has been essential to convert raw sequencing data into biologically meaningful information. To perform microbiome analysis from the data generated by amplicon sequencing and metagenomics, it is necessary to have bioinformatics knowledge (Table 4). Microbiome data analysis involves software that requires familiarity with the Shell environment and programming languages such as R and Python [46].

To analyze reads from 16S gene amplicons generated with Illumina, it is a common practice to use tools such as USEARCH [158] or QIIME 2 [159,160]. These packages have most of the bioinformatics tools needed for microbiome analysis. In the case of metagenomic studies performed with Illumina technology, MetaPhlAn2 [161] or kraken2 [162] can be used for taxonomic classification, and MEGAHIT [163] or metaSPAdes [164] for the assembly of the reads.

Different tools are available to work with data generated with PacBio, depending on the type of analysis. If the objective is to analyze the full-length 16S gene data, the following can be used: DADA2 [130], QIIME 2 [159,160], microbiome helper [165], OneCodex [166], EZBiome [167], or 16S PacBio GitHub pipeline [168]. Taxonomic and functional classification can be performed in metagenomics with the PacBio GitHub pipeline [169] or BugSeq [170]. For *de novo* assembly of complete or near-complete metagenomes from metagenomic data, Hifiasm-Meta [100], metaFlye [171], or metaMDBG [172] together with the PacBio GitHub pipeline [168] can be used.

In the case of ONT, amplicon and metagenomics data can be easily analyzed with EPI2ME^TM^, a platform compatible with Windows, macOS, and Linux. EPI2ME enables data analysis for all levels of expertise, which has changed the bioinformatics paradigm by allowing anyone to analyze their data. EPI2ME is an intuitive platform with different workflows that are continuously curated and updated, such as wf-16S (taxonomic classification of 16S amplicons), wf-metagenomics (taxonomic classification of individual shotgun metagenomic reads), or wf-bacterial-genomes (assembly of bacterial genomes). Each workflow generates an interactive report in HTML format and can be run locally, in the cloud, or via the command line, adapting to different levels of bioinformatics expertise [173]. In addition, ONT and its scientific community offer tools on GitHub for advanced users [174]. These include tools used to increase the accuracy of ONT 16S sequencing data, such as NanoClust [175].

It is worth noting that there are tools that work with data obtained from different sequencing platforms, which makes them interesting for optimizing downstream analysis and comparative studies. For example, metaFlye [171] and metaMDBG [172] are designed to assemble long and accurate metagenomic reads from both PacBIO HiFi and ONT. Platforms such as OneCodex [166], Emu [176], and BugSeq [170] work with data from both Illumina, PacBio, and ONT, which increases their versatility.

### 7.8. General Comparison

A comparison of the sequencing platforms based on the normalization of seven key characteristics is represented in Figure 5a: long-read length (bp), accuracy, runtime (hours), total output (Gb/run), instrument price ($), instrument portability, and bioinformatics expertise.

The best value for each characteristic was assigned with a score of 1, and the rest received proportional scores. Specific scales were established for qualitative characteristics such as instrument portability and bioinformatic expertise, ranging from 0 to 1. For instrument portability, a score of 1 indicates a portable system, 0.67 indicates a compact benchtop system, 0.33 indicates a desktop system, and 0 indicates a production-scale system. For bioinformatics expertise required for data analysis, a score of 1 was assigned to platforms suitable for all levels of expertise (beginner to advanced), 0.5 to intermediate/advanced platforms, and 0 to advanced platforms.

In addition, Figure 5b represents a ranking of the sequencing platforms ordered by the sum of the scores obtained for each desirable characteristic.

After evaluating the desirable features for all the long-read sequencing platforms from ONT and PacBio and commonly used Illumina platforms, ONT platforms were the first in the ranking, with Flongle and MinION at the top. Therefore, the following section discusses the unique applications of these platforms due to their outstanding features.

## 8. ONT Applications: Portable, Affordable, Fast, and Real-Time Sequencing

As discussed above, each sequencing platform has advantages and disadvantages that must be evaluated according to the objectives and requirements of each research study. In this context, ONT has introduced some unique features that have brought about an unprecedented revolution in the investigation of the genetic composition of microbial communities. These features include low cost, portability with minimal power requirements, the ability to obtain fast and real-time results, and the possibility of adaptive sampling [107].

Legacy short-read sequencing platforms are often large, require professional precision calibration, are complicated to transport, and depend on power infrastructure. As a result, they are centralized in well-resourced locations, often delaying the time to results. Against this, ONT provides the only portable sequencing platform, MinION, which is not restricted to a laboratory environment and can be transported directly to the sample site. Using MinION saves significant time and minimizes the risk of sample degradation, resulting in a more accurate representation of microbial diversity. A striking example is the study by Gowers et al., which analyzed ice sheet microbial communities by MinION sequencing using only solar energy, completely off-grid, on an 11-day ski and sledge trek across Iceland [177].

Another interesting example of the potential of the portability of MinION is that it has been used to sequence genomic DNA extracted from the *Enterobacteria lambda phage*, *Escherichia coli*, and *Mus musculus* on the ISS. The ISS presents extreme conditions in a free-fall and constant-microgravity environment, orbiting 400 km above the Earth and travelling at 28,000 km/h [178].

MinION was also integrated for the first time as part of a university curriculum, where students collected water samples from aboard a research vessel that sailed for 7 days in the Bering Sea. Despite the adverse weather conditions, the students could perform real-time sequencing of the DNA extracted from the samples [179].

In contexts where urgent action is needed to assess the state of biodiversity to design effective conservation plans, ONT technology can be a key tool. For example, Madagascar’s biodiversity is threatened by deforestation, habitat destruction, and poverty. Implementing portable genetic technologies, such as a mobile laboratory equipped with a miniaturized thermal cycler and an ONT sequencer, enabled a rapid assessment of local biodiversity in a Reserve. This strategy provided immediate conservation results and trained local scientists, strengthening their role in environmental management [180].

In terms of rapid diagnostic potential, ONT is unrivalled. Shortened protocols for DNA extraction, library preparation, and high-speed sequencing have been developed, providing a unique ability to achieve diagnostics in reduced timescales [6]. For example, when using metagenomics with MinION, fungal pathogens were identified from samples of conifers (Pinaceae) and potatoes (*Solanum tuberosum*) in less than 150 min after sample collection [181].

In situations where infections are life-threatening, the time it takes to identify a pathogenic microorganism accurately is crucial. A proof-of-concept study illustrated the potential of nanopore metagenomics to generate high-accuracy real-time data identifying pathogens and antibiotic resistance genes in critical healthcare settings. They achieved results in an average of 6.7 h from lower respiratory tract sampling, significantly faster than results obtained with traditional culture-based techniques that required an average of 40 h. In addition, this methodology allowed the identification of hidden infectious loads in intensive care units that could not be detected by routine testing because they were unexpected and/or non-culturable [182].

Utilizing ONT technology can also offer a valuable strategy due to its rapid, real-time results in investigating foodborne outbreaks. In the European Union, the responsible food source cannot be determined in up to 60% of reported foodborne outbreaks. This deficiency is mainly due to the lack of food debris and the bias that occurs with current detection procedures due to the need for enrichment to achieve detectable bacterial abundances in the sample. Recently, a study was conducted combining the adaptive sequencing (targeted sequencing) and metagenomic capabilities of ONT to avoid the need for culture enrichment. This method showed great potential for the rapid, accurate characterization of pathogens at the strain level without the need for culture, contributing to improved food safety and public health [183].

## 9. Additional Key Aspects When Using Long-Read Sequencing for Microbiome Analysis in Ecosystems

Some key aspects that should be considered in microbiome analysis are well-established for short-read sequencing platforms but remain less defined for long-read platforms. These aspects include the choice of amplicon or shotgun sequencing approach, the sequencing depth to be achieved, and the availability of databases for microbiome data. While most references and established protocols for these considerations are addressed to short-read sequencing methods, the increase in the use and competitiveness of long reads highlights the need for new perspectives. Therefore, this section discusses the current status of these aspects, to provide preliminary guidance for the effective use of long-read sequencing technologies in microbiome analysis.

### 9.1. Current Perspectives of Amplicon and Shotgun Sequencing Approach

Before conducting microbiome research, researchers need to consider the objective of the work and define the aspect they aim to investigate. These considerations are determinants for establishing the logistical requirements and costs and the need for amplicon or shotgun sequencing as well as multidisciplinary approaches [184].

#### 9.1.1. 16S Ribosomal RNA (16S) Gene Amplicons

Over the last 25 years, amplicon-based/marker gene sequencing has been the most widely used method for analyzing microbial community composition through various samples and treatments [185,186]. Several specific marker genes have been identified and are widely used for amplicon sequencing in bacteria, archaea, and fungi. Many marker genes are functionally conserved across phylogenetic distances, allowing them to function as molecular clocks for investigating evolutionary transitions and changes [127].

The major marker gene employed in prokaryotes for amplicon sequencing is 16S, considered the gold standard in microbial typing, offering the great advantage of selecting only bacterial and archaeal DNA [127,187].

The 16S gene, due to its high conservation, plays an essential role in cell function and survival, making it a fundamental tool for classifying known and unknown microbial taxa [127]. Direct sequencing of 16S gene amplicons allows the analysis of phylogeny, taxonomy, and the abundance of species or taxonomic groups in a microbiome. It is also known as the massively parallel sequencing of partial 16S gene amplicons because it allows the sequencing of multiple reads simultaneously [46].

The relatively short size of the 16S gene (~1542 bp) facilitates sequencing, even in large samples. The gene sequence includes highly conserved primer binding sites and nine variable regions (V1–V9) [127]. The first step in 16S gene amplicon sequencing involves PCR amplifying full-length or partial 16S genes using primers recognizing conserved regions. The amplicons are then sequenced, and the sequences are taxonomically identified by comparison against a reference database [18].

It should be noted that a wide choice of PCR primers for the 16S gene is currently available; each presents advantages and disadvantages (Table 5). Optimal primers should reduce amplification bias and amplify a region that provides taxonomically and phylogenetically useful information, depending on the analyses performed. In addition, for a good selection of primers, it is important to consider the expected composition of the microbial community to be analyzed [18]. It is also recommended to choose primers used in similar published articles [46].

Amplification of the V1–V2/V3, V3–V4/V5, or V4 regions is commonly employed in microbiome community analysis [188] (Table 5). In the HMP, primers were mostly used to amplify the V1–V3 or V3–V5 regions [189]. For example, amplification targeting the V1–V2 regions is widely used because it is highly specific to bacteria (not archaea and eukaryotes). However, this region performed poorly in classifying sequences for the *Bifidobacterium* genus [190] and the Verrucomicrobia [191] and Proteobacteria phyla [53]. The V3–V5 region performed poorly in classifying sequences from the phylum Actinobacteria but was good for *Klebsiella.* The V1–V3 region performed well for *Escherichia/Shigella* and could provide information at the species level [53]. The V3–V4 region failed to detect Chloroflexi and Elusimicrobia phyla [192]. The V4–V5 region showed low coverage of Bacteroidota and produced few overlaps with other primer pairs [188]. The V6-V9 region performed well in classifying the genera *Clostridium* and *Staphylococcus* [53]. For archaeal profiling in complex microbial communities, V1–V2, V3, and also V4–V5 regions have been targeted [127,193,194].

Even short fragments (as small as 100 bp) have revealed changes in microbial community composition [195]. However, the taxonomic resolution achieved using rRNA regions is much lower than that achieved using the full-length 16S gene, the internal transcribed spacer (ITS) region, or the 23S rRNA gene. The best resolution at the strain and species levels was achieved by combining the ITS regions with the 16S or 23S rRNA gene (or both), but it requires long-read sequencing technologies. Amplification and sequencing of the 16S-ITS-23S region (~4500 bp) has produced good results for distinguishing between *Escherichia coli* and *Shigella* spp. (Table 5). However, the length of the ITS region is highly variable, which can lead to PCR and sequencing biases. In addition, the rRNA genes are not positioned conventionally in many cases, and taxa can be lost when targeting the long ITS region fragments [6,196].

Although the taxonomic accuracy achieved by sequencing the entire 16S gene cannot be achieved using regions, most microbiome studies available amplify and sequence only a part of the 16S gene. This bias is mainly due to the extensive use of sequencing platforms such as Illumina, whose technology limits the read length to 300 bp [53,197].

As sequencers become more powerful, researchers incorporate barcode sequences in PCR primers to identify each sample and sequence several samples simultaneously [189]. As many sequences are read, bacteria at low relative abundances can also be detected [198]. Studies that sequence the 16S gene typically collect around 10,000 sequences per sample to estimate microbial species abundance [159].

Amplicon sequencing offers advantages such as reliable taxonomic identification, unknown bacteria identification, high velocity, and the capacity to gather quantitative data [195]. It is low-cost and fast compared to metagenomics shotgun sequencing methods [184]. In addition, the sequencing data are not as complex, facilitating their analysis. It can be used with low-biomass specimens and host DNA-contaminated samples [46].

Despite the power of amplicon sequencing, it also has limitations [199]. First, failures in diversity resolution may occur due to biases in DNA extraction and PCR [200,201]. Second, there are discrepancies in the selection of PCR primers or hypervariable regions for achieving a higher taxonomic resolution [202] and in the taxa 16S gene copy number [203]. Third, amplicon sequencing commonly provides information on the taxonomic composition of the microbial community but cannot directly determine its biological functions. Sometimes, the functions encoded by a genome are inferred from the particular 16S sequence it contains, so achieving an accurate estimate depends on a reliable taxonomic identification of the community [87]. Finally, amplicon sequencing is used to analyze those organisms for which amplifiable taxonomic markers are known [199]. Additionally, horizontal transfer of the 16S gene between distant taxa is possible, which may lead to erroneous estimates of community composition [204].
microorganisms-13-01861-t005_Table 5Table 5Comparison of the characteristics of commonly targeted regions for taxonomic profiling in microbial community analysis: rRNA operon, full-length 16S rRNA gene, and 16S rRNA gene hypervariable regions (V1–V2, V1–V3, V3, V4, V3–V4, V3–V5, V4–V5, and V6–V9).Target RegionPrimer PairsAmplicon Length (~)PrimerSpecificityAccurate TaxonomicResolutionOther Remarks16S-ITS-23SrRNA operon27F, 519F, 2241R and 2428R [6,196]4500 bp [196]Universal[6,196]Species and strain levels [6,196]It is especially useful for distinguishing *Escherichia coli* and *Shigella* spp.; limitations in detecting archaeal taxa; emerging method; requires long-read sequencing [6,196]V1–V9 (16S rRNA)27F-1492R [6]1465 bp [53]Universal[188]Species and strain levels [53,205]Better taxonomic resolution than 16S regions; 27 F primer has limited amplification for *Bifidobacterium* [206]; requires long-read sequencing [53]V1–V227F-338R [207]310 bp [53]Universal [188]Genus level; good for archaea [127,193]Low sensitivity for *Bifidobacterium* [190], Verrucomicrobia [191], and Proteobacteria [53]; suitable for low-bacterial biomass samples [208]; recommended region for sputum microbiome analysis; commonly used with Illumina [202]V1–V327F-534R [209]507 bp [53]Universal [188]Genus level; informative at species level [53]Good sensitivity for *Escherichia/Shigella.* Poor for *Bacteroides intestinalis* [53] and Verrucomicrobia [191]; used in HMP (454) [189]; recommended region for plant [210] and skin microbiome analyses [211]; suitable for long-read sequencing platformsV3338F-533R [212];ARC344F-519R [193,213]200 bp[212,213]Bacteria: 338F-533R [212]; archaea: ARC344F-519R [193,213]Genus level; ARC344F-519R good for archaea [193,212,213]ARC344F-519R is considered the best choice for archaea community profiling [193]V4515F-806R [214]291 bp [53]Universal [188]Genus level [53]Susceptible to human DNA amplification [208]; recommended region for diverse microbial communities [18]; used in EMP; commonly used with Illumina [215]; reduced bias against the SAR11 bacterial clade with 806RB primer [216]V3–V4341F-785R [192]464 bp [192]Bacteria [192]Genus level [192]Fails to detect Chloroflexi and Elusimicrobia [192]; widely used region for human-associated, soil, and plant microbiome analysis; commonly used with Illumina [188]V3–V5357F-926R [53]569 bp [53]Bacteria [188]Genus level [53]Susceptible to human DNA amplification [208]; good sensitivity for *Klebsiella* and poor for Actinobacteria [53]; used in HMP (454) [189] and MetaHit [217]; suitable for long-read sequencing platformsV4–V5515F-944R [188] 429 bp [188]Bacteria [188]; 515F-Y/926R universal [194]Genus level [218]; 515F-Y/926R good for archaea [194]Low sensitivity for Bacteroidota, with few overlaps with other primer pairs [188]; 515F-Y/926R primer pair has reduced bias against environmental archaea *Crenarchaeota*/*Thaumarchaeota* [194]; 515F-Y/926R is widely used in marine microbiome studies and tested in temperate water microbiomes [194,219]V6–V9968F/1492R [53]524 bp [53]Bacteria [188]Genus level [53]Good sensitivity for *Clostridium* and *Staphylococcus* [53]; suitable for long-read sequencing platforms EMP, Earth Microbiome Project; HMP, Human Microbiome Project; MetaHit, Metagenomics of the Human Intestinal Tract.

#### 9.1.2. Metagenomic Shotgun Sequencing

A shotgun metagenomics approach refers to the unselective or shotgun sequencing of all (meta-) the microbial genomes (-genomics) found in a given sample [72]. This approach stands out because all DNA fragments in a sample are sequenced instead of specific fragments [220]. The metagenomic approach differs from the amplification approach because the latter involves a PCR amplification step for region-specific amplification. Shotgun sequencing allows for taxonomic and functional profiling of microbial communities and the reconstruction of partial or full genome sequences [72]. It is also known as massively parallel sequencing of the whole genome because it allows the sequencing of multiple reads simultaneously [46]. Sequencing of the 16S gene, which targets specific organisms or marker genes, is sometimes called metagenomics. However, using this term in this context is incorrect since it does not address the whole genomic content present in a sample [72].

Microbiome shotgun metagenomics sequencing entails the random fragmentation of whole community DNA and massively parallel sequencing of DNA fragments [195]. These sequences (reads) can be aligned to diverse genomic locations in numerous genomes within the sample. Some of these sequences are derived from taxonomically informative genomic sites (such as the 16S gene), allowing for the sample’s taxonomic profiling. Other sequences originate from coding regions, which allows for biological function profiling. Shotgun metagenomics allows us to simultaneously address the questions, who is in the microbial community, and what are they doing there? [199]. Moreover, overlapping sequences can be computationally assembled to reconstruct full or partial genomes [184,221].

Metagenomics was first described for microbial populations by Handelsman in 1998 when analyzing an unknown soil microbiome [222]. In 2003, the first description of the metagenomics of the gastrointestinal tract was carried out with the analysis of the uncultured viral community present in human feces, leading to an estimated 1200 recognizable viral genotypes [223]. In 2006, humans were described as super-organisms in terms of their genes and metabolites, as they include not only those inherent to humans but also those related to the associated microbial community. Metagenomics was used to demonstrate that the microbiome is enriched with key genes essential to humans involved in the metabolism of glycans, amino acids, or biosynthesis of vitamins, among others [224].

Since then, the metagenomics approach has been employed in numerous microbiome studies, from the HMP [26] to the analysis of microbial populations in seawater samples from the Sargasso Sea expedition [225].

Microbiome shotgun metagenomics has the advantage of collecting data on the genetic diversity and functionality of the microbial community, distinguishing it from other techniques that only analyze genetic diversity [195]. The functional potential of a microbial community can be studied indirectly by utilizing marker gene approaches or directly by functional gene analysis and related pathways by metagenomic shotgun sequencing [184]. Although this technique avoids biases related to amplification and resolution, it has its own methodological and computational biases and limitations [184].

First, metagenomics analysis is technically challenging, and complex and large data generate difficulties in computational analysis [195]. For example, the metagenome of a microbial community is very diverse, so it is not easy to produce a complete representation of that genome in reads [199]. Finally, identifying significant portions of genomes for all the species often requires a large volume of data, which can lead to computational challenges due to the extensive genomic information within samples. Because of that, new informatics software is being developed to enhance the simplicity and efficiency of metagenomic data analysis [199].

Second, metagenomes might include an undesired host or non-target DNA, particularly in microbiome research. When host DNA overwhelms microbial DNA, specific techniques are necessary before sequencing to enrich microbial DNA selectively [226]. Bioinformatic methods have also been developed to filter host DNA [199].

Third, detecting and removing contamination in metagenomics samples is especially challenging [227]. Some tools have been developed to identify and eliminate contaminants in metagenomic sequences [228].

Finally, metagenomics has a higher cost than amplicon sequencing, particularly in complex communities or when the amount of host DNA significantly exceeds microbial DNA [199]. Shallow shotgun metagenomics has a cost per sample comparable to amplicon sequencing and allows for obtaining taxonomic profiles but not performing functional profiling or genome reassembly due to the lack of coverage [229].

### 9.2. Current Perspectives on Sequencing Depth

Sequencing depth refers to the number of sequencing reads generated per sample. This parameter varies depending on the technology used, the type of sample, the number of samples multiplexed, and the study’s objective. Working with appropriate sequencing depths is essential to achieve the correct microbiome characterization, as it affects the sensitivity and specificity of the analysis [230]. No universal “standard” depth has been described, and recommendations should be tailored to the design of the experiment.

Shallow metagenomic sequencing is commonly used for microbial community profiling as it has a higher taxonomic resolution capacity than short-read 16S sequencing and is much cheaper than deep metagenomic sequencing [99,229,231]. When using short-read sequencing technologies, such as the Illumina platform, shallow metagenomic sequencing is defined as 2–5 million (M) reads per sample, while deep metagenomic sequencing exceeds 10 M reads per sample [231]. On the other hand, ultra-deep metagenomic sequencing, with more than 20–60 M reads per sample, is required to detect rare taxa and, in specific cases, to recover high-quality genomes from communities expected to have a high proportion of new microbial species [99,232].

In one study, Illumina HiSeq metagenomics sequencing was performed on fecal samples, and the microbial taxonomic classification results were analyzed with different methods for sequencing depths of 5, 10, 20, 40, 80, and 100 M read pairs. They concluded that the taxonomic resolution did not improve above 60 M read pairs [233]. It should be noted that the read depth primarily improves sensitivity rather than taxonomic specificity, and that this conclusion is highly dependent on the complexity of the sample [73].

Studies that sequence the specific regions of the 16S gene typically work with around 10,000–15,000 sequences per sample when you want to estimate relative microbial abundance [159,234,235]. The Earth Microbiome Project (EMP) rarefied 5000 sequences of the V4 region of the 16S gene per sample [215]. However, even fewer reads, around 2000, may be sufficient to obtain basic microbial profiles [214].

With PacBio’s technology, it is estimated that when using the standard protocol for the complete 16S gene, up to 384 samples can be obtained with about 10,000 (Vega^TM^ system) or 20,000 reads (Revio^®^ system). If the Kinnex kit is used, up to 1152 samples (Vega) with 30,000 reads per sample or 1536 samples (Revio) with 45,000 reads per sample can be analyzed [236]. A recent study compared the taxonomic resolution of 16S gene sequencing with Illumina’s short read and PacBio’s long read using microbiome samples from saliva, subgingival plaque, and feces. In the study, they worked with an average of 12,500 reads per sample in PacBio (average length 1457 bp) and about 90,000 reads in Illumina (average length 414 bp) [134].

For metagenome profiling, up to 64 samples can be analyzed with ~0.75 Gb per sample (Vega) or 128 samples (Revio). In the case of metagenomic assembly, up to 8 samples can be analyzed with ~7 Gb per sample (Vega) or 16 samples with ~6 Gb per sample (Revio) [236].

When working with ONT, sequencing the complete 16S gene is also considered a good starting point for sequencing depths of 10,000 reads, although 20,000–40,000 reads per sample are common [4,131,237].

### 9.3. The Emergence of Microbiome Databases Specific to Ecosystems

The exponential increase in heterogeneous data obtained from various protocols for collecting, processing, and analyzing microbiome samples, with or without standardization, has led to the development of databases that seek to create more consistent and specific repositories. Developing generic microbiome databases has been a milestone in standardizing protocols and data accessibility from laboratory studies worldwide. These databases are usually grouped by data type (such as metabarcoding, metagenomics, metatranscriptomics, metaproteomics, metabolomics, or physiological data) and target organisms [238].

An emerging approach is the development of ecosystem-specific databases (ES-DBs). ES-DBs aim to standardize microbiome sample methodologies and analyses based on the unique ecosystem characteristics from which the sample originates. ES-DBs would facilitate the interconnectivity of spatially and temporally distinct microbiome studies conducted in the same ecosystem to obtain a coherent and detailed view of how a microbial community interacts within its ecosystem. A great example of these ES-DBs would be the Microbial Database for Activated Sludge (MIDAS 3) [239], developed specifically for wastewater treatment systems and with a taxonomic resolution at the species level [238].

For a database to be of high quality, it requires good datasets. As technological advances occur, updates to the standard methodology become necessary. Database curators decide when and how to incorporate these updates, considering that each change affects the reproducibility of the database. For example, the current strategy the EMP recommends is short-read sequencing of the 16S gene. However, an improved accuracy and reduced costs have made long-read sequencing methodologies increasingly competitive in microbiome research. Therefore, each database must establish criteria to continue incorporating new datasets without losing a high quality [238].

## 10. Conclusions

The sequencing platforms used to study microbial communities in different ecosystems have evolved significantly in recent years. The shift from culture to Sanger sequencing made it possible to increase the microbial diversity detectable in samples. Subsequently, the advent of NGS, first with 454 and then Illumina, brought about a real revolution in microbiome study. Illumina enabled the generation of fundamental insights into microbial ecology and function [28]. However, this platform only sequenced short-read fragments compensated with a high depth and low cost, resulting in a taxonomic resolution setback. Long-read sequencing platforms like PacBio and ONT were developed to overcome this limitation. Initially, these platforms had a lower basecall accuracy compared to Illumina. However, in recent years, these platforms have incorporated strategies that have raised their accuracy to competitive levels [6].

Long-read sequencing platforms offer unique insights into the conventional analysis of microbial communities in ecosystems. These technologies allow for a more accurate taxonomic resolution for bacteria and fungi [87]. Moreover, they can capture complete viral genomes, which promotes further virome research [76,77,78]. In addition, they have revolutionized the contiguity and accuracy of MAG assembly by allowing sequencing of fragments larger than 10 kb [99]. Another highlight of these platforms is the ability to detect epigenetic modifications directly in microbial genomes [80,81,121].

Selecting the sequencing platform that best suits the needs of each project can be complex. To facilitate this choice, a selection of seven key features for microbiome sequencing methods is proposed: long-read length (bp), accuracy, runtime (hours), sequencing output per cell (Gb/run), accessibility, portability, and bioinformatic expertise required. These characteristics were evaluated on the three most commonly used sequencing platforms for microbiome analysis: Illumina, PacBio, and ONT. Flongle and MinION (both from ONT) were the platforms that ranked highest, thus highlighting some unique applications for the study of microbiomes in ecosystems.

ONT technology’s portability, low cost, and real-time sequencing have facilitated in situ studies in extreme or difficult-to-access environments, such as glaciers in Iceland [177], the International Space Station (ISS) [178], or research ships at sea during storms [179]. It has also been essential in resource-constrained environments, such as protecting biodiversity in Madagascar [180] or controlling outbreaks, as with the 2015 Ebola outbreak in West Africa [157], opening up new opportunities for real-time environmental and clinical microbiology.

Finally, some additional key aspects of microbiome analysis that are well-established for short-read sequencing platforms but not yet standardized for long-read platforms have been discussed. These aspects include choosing between amplicons and the shotgun approach, which depends on the study’s objectives [184]. Another aspect is the required sequencing depth, for which no universal standards exist. For 16S gene amplicon analysis, 10,000 reads per sample is generally considered a good starting point, whereas metagenomics depends on whether profiling or assembly is desired. The availability of ES-DB for microbiome data could also be important. Although these databases have focused on short readings, the increasing use and competitiveness of long reads mean they should adapt to include these data [238].

Overall, this review provides a guide to the evolution of sequencing methods and a practical tool for selecting the correct sequencing approach in each microbiome study.

## Figures and Tables

**Figure 1 microorganisms-13-01861-f001:**
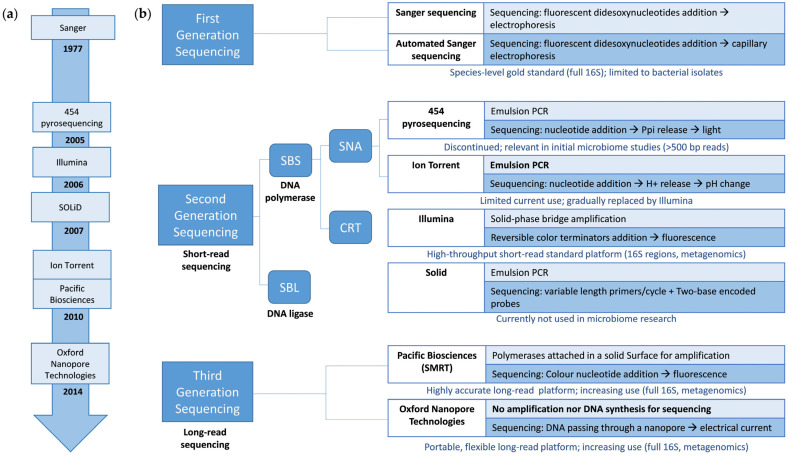
(**a**) Timeline with the introduction of sequencing technologies employed in microbiome research. (**b**) Schematic classification of first-, second-, and third-generation sequencing technologies employed in microbiome research. The key sequencing points for each sequencing technology amplification strategy are indicated (if utilized). In addition, the current use of each technology in microbiome research is provided below each technology. Abbreviations: bp, base pair; CRT, cyclic reversible termination; SBS, sequencing by synthesis; SBL, sequencing by ligation; SNA, single-nucleotide addition; SMRT, single-molecule real-time sequencing.

**Figure 2 microorganisms-13-01861-f002:**
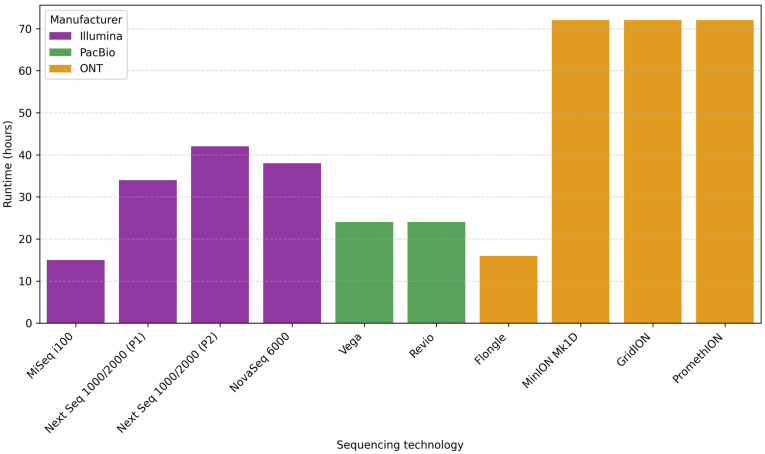
Comparison runtime (hours) for different ONT, PacBio, and Illumina sequencing platforms. Abbreviations: ONT, Oxford Nanopore Technologies; PacBio, Pacific Biosciences.

**Figure 3 microorganisms-13-01861-f003:**
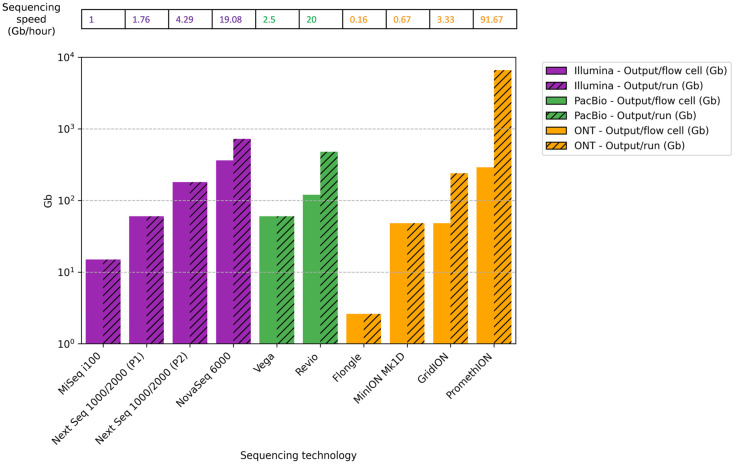
Comparison of maximum theoretical output per cell (Gb), total output/run (Gb), and sequencing speed (Gb/hour) across different ONT, PacBio, and Illumina sequencing platforms. Abbreviations: ONT, Oxford Nanopore Technologies; PacBio, Pacific Biosciences; P1, P1 flow cell; P2, P2 flow cell.

**Figure 4 microorganisms-13-01861-f004:**
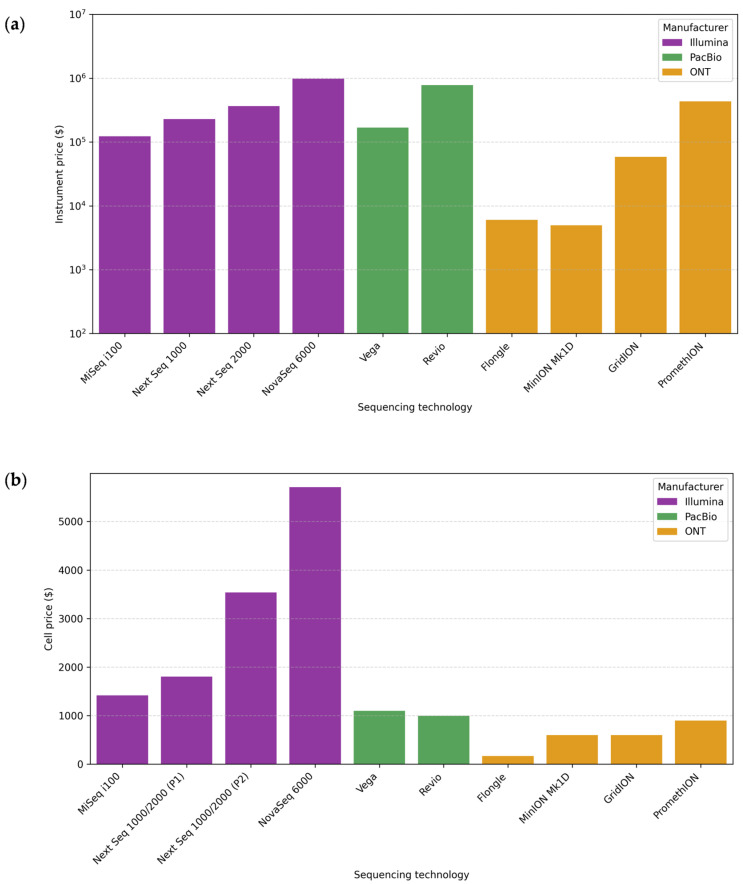
Comparison of sequencing instrument and cell prices for different ONT, PacBio, and Illumina sequencing platforms. (**a**) Instrument price ($). (**b**) Cell price ($). Abbreviations: ONT, Oxford Nanopore Technologies; PacBio, Pacific Biosciences; P1, P1 flow cell; P2, P2 flow cell.

**Figure 5 microorganisms-13-01861-f005:**
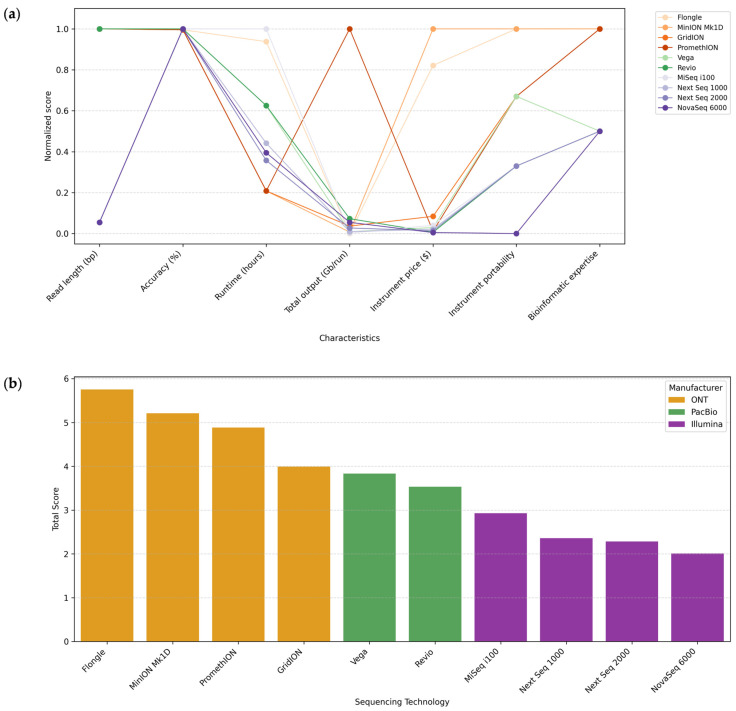
Integrated overview of characteristics for microbiome analysis in ecosystems of commonly used sequencing platforms: ONT, PacBio, and Illumina. (**a**) Comparison of desirable characteristics among the different sequencing platforms: read length (bp), accuracy, runtime (hours), total output (Gb/run), instrument price ($), instrument portability, and bioinformatics expertise. (**b**) Global ranking of the different sequencing platforms, calculated by summing the scores assigned to each desirable characteristic.

**Table 1 microorganisms-13-01861-t001:** Comparison of 16S and metagenomics read length and applications for different sequencing platforms.

Sequencing Platform	Maximum 16S Read Length	Taxonomic Resolution (16S)	Metagenomics Read Length	Metagenomic Applications
Illumina	2 × 300 bp (overlap ~50 bp)	Mainly genus level	2 × 300 bp (overlap ~50 bp)	Assembly (fragmented), taxonomic and functional profiling
PacBio	~1500 bp	Species and strain levels	~10 kb	High-quality assembly, taxonomic and functional profiling
ONT	~1500 bp	Species and strain levels	~10 kb	High-quality assembly, taxonomic and functional profiling

Abbreviations: ONT, Oxford Nanopore Technologies; PacBio, Pacific Biosciences.

**Table 2 microorganisms-13-01861-t002:** Overview of the initial sequencing errors, the strategies used to achieve higher accuracy, and the current accuracy of Illumina, PacBio, and ONT sequencing platforms.

Sequencing Platform	Initial Error Source	Initial Error Type (%)	Accuracy Improvement Strategies	Current Accuracy
Illumina	Library construction Sequencing process DNA damage [141]	Substitutions (after homopolymer, G/C > A/T; ~0.01–0.5%) [142]	XLEAP-SBS chemistry [143]	~99.9% (≥85% of bases) [144]
PacBio	Fluorescence signals’ misinterpretation Polymerase errors [145]	Substitutions (A↔C, G↔T; ~1.7%) Deletions (~3.2%) Insertions (~8%) [145]	HiFi reads (CCS) [146]	~99.9% (0.5–5 kb; 95% of bases; 10–15 kb; 90% of bases) [147]
ONT	Nanopore design leads to bias in homopolymers (A/T) [145]	Substitutions (A↔G, C↔T; ~4%) Deletions (~4%) Insertions (~4%) [145]	Kit 14 chemistry R10.4.1 flow cell Basecaller updates Duplex reads [68]	~99.9% (duplex reads) ~99% (simplex reads) [68,69]

Abbreviations: CCS; circular consensus sequencing; HiFi, high fidelity; ONT, Oxford Nanopore Technologies; PacBio, Pacific Biosciences.

**Table 3 microorganisms-13-01861-t003:** Instrument portability and size specifications for different ONT, PacBio, and Illumina sequencing platforms.

Sequencer	Manufacturer	Portability	Size
MiSeq i100	Illumina	No	Benchtop
Next Seq 1000/2000	Illumina	No	Benchtop
NovaSeq 6000	Illumina	No	Production-scale
Vega	PacBio	No	Compact benchtop
Revio	PacBio	No	Benchtop
Flongle	ONT	Yes	Palm-sized
MinION Mk1D	ONT	Yes	Palm-sized
GridION	ONT	No	Compact benchtop
PromethION	ONT	No	Compact benchtop

Abbreviations: ONT, Oxford Nanopore Technologies; PacBio, Pacific Biosciences.

**Table 4 microorganisms-13-01861-t004:** Bioinformatic expertise requirements to analyze data generated by ONT, PacBio, and Illumina sequencing platforms.

Sequencing Platform	Bioinformatic Expertise
Illumina	Required (intermediate/advanced)
PacBio	Required (intermediate/advanced)
ONT	User-friendly tools (beginner to advanced)

Abbreviations: ONT, Oxford Nanopore Technologies; PacBio, Pacific Biosciences.

## Data Availability

No new data were created or analyzed in this study. Data sharing is not applicable to this article.

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
