# Peer review of "Why Are Long-Read Sequencing Methods Revolutionizing Microbiome Analysis?"

_microorganisms, 2025, doi:10.3390/microorganisms13081861_

Round 1
Reviewer 1 Report
Comments and Suggestions for Authors
The authors' manuscript, if well-intentioned and well-polished, can contribute to science. My comments are submitted as attachments.

Author Response
Manuscript ID: microorganisms-3742685
Response to Reviewer 1
Thank you for your contribution and revision of the manuscript. Your suggestions are important and bring an improvement to the work's content. We include the answers after each comment.
Comment 0: The authors explain the advantages of long-read sequencing by comparing different sequencing platforms and comparing primers that amplify the 16s rRNA region used in different microbiomes. Although long-read sequencing has some disadvantages in terms of cost, it is a technology that has clear advantages, and the content recommending it will contribute sufficiently to the advancement of science. I would like to express my gratitude to the authors for introducing content that can contribute to science, and I would like to offer my opinion after reading the manuscript.
Thank you for your words and the revision of the manuscript.
Comment 1: Why long-read sequencing methods are revolutionizing microbiome analysis in ecosystems? Although the word revolution may seem a bit exaggerated, it seems like a title that will grab the readers' attention.
Thank you for your comment.
Comment 2: Finally, amplicon and metagenomics approaches and sequencing depth are discussed when using long-red sequencing technologies in microbiome studies. In conclusion, the sequencing method used for the microbiome in ecosystems study should ideally meet the following requirements: long-read length, high accuracy, short runtime, high output, accessibility, portability and flexible bioinformatic expertise. Although no single method possesses all features, long-read sequencing represents an advance over conventional short-read methods. The abstract part is well written overall, but it is not clear what the authors are ultimately trying to say. The authors state that long-read sequencing is an advanced form of short-read sequencing, which provides too limited information to the readers. There should be a mention of the subject of the advancement, so that the readers can understand what has advanced from short-read sequencing to long-read sequencing and why long-read sequencing is good to use. For example, it would be good to briefly mention the advantages and current limitations of long-read sequencing, such as "long-read sequencing has higher accuracy than short-read sequencing, so it is recommended to accumulate more research cases in the future."
You are right. We have included the following sentence [page number 1, Abstract, and lines 26-27]: In conclusion, although no single sequencing method currently possesses all the ideal features for microbiome analysis in ecosystems, long-read sequencing technologies represent an advancement in key aspects, including longer read lengths, higher accuracy, shorter runtimes, higher output, more affordable costs, and greater portability. Therefore, more research using long-read sequencing is recommended to strengthen its application in microbiome analysis.
Comment 3: Microbiome, microbiota. The authors use the words microbiota and microbiome hundreds of times, but there are sentences where the scope of the term is not appropriate. Please construct sentences in a professional manner that covers the scope of the definition of the term.
Thank you, we have revised the text for better coherence.
Comment 4: This review aims to 1) analyze the historical evolution of sequencing technologies, their advantages and disadvantages for microbiome analysis, 2) elaborate a proposal for desirable characteristics of microbiome sequencing methods to facilitate the choice according to the study's objectives, and 3) to provide an informative and visual comparison of the main platforms used in microbiome analysis. Finally, other important points are addressed when using long-red sequencing technologies in microbiome studies, such as amplicon, metagenomics approaches, and sequencing depth. The sentences in this paragraph are too general. 1), 2), and 3) are ultimately intended to highlight the advantages of long-read sequencing. 1), 2), and 3) talk about general content, and then suddenly narrow down to long-read sequencing in the last sentence, which confuses the reader's flow of thought. Please reorganize the paragraph content so that it is clear that 1), 2), and 3) are intended to highlight the advantages of long-read sequencing.
Thank you for your comment. We have included the following explanatory sentences:
- [page number 2, paragraph 6, lines 69-76]: This review aims to explore the potential advantages of long-read sequencing in microbiome of ecosystems studies by: 1) analyzing the historical evolution of sequenc-ing technologies, their advantages and disadvantages for microbiome analysis; 2) elab-orating a proposal for desirable characteristics of microbiome sequencing methods to facilitate the choice according to the study's objectives; and 3) providing an informative and visual comparison of the main platforms used in microbiome analysis. Finally, other important points are addressed, including amplicon and metagenomics approaches, as well as sequencing depth.
Comment 5: 4.1.1. 454 pyrosequencing. 4.1.2. Ion Torrent. The authors introduce that researchers using this platform amplified the V1-V3 and V3-V5 regions in [4.1.1. 454 pyrosequencing], but the explanation is insufficient when introducing [4.1.2. Ion Torrent]. There are many previous cases of amplifying the V3-V4 and V4-V5 regions using Ion Torrent. In addition, the Illumina system is described in great detail, but Ion Torrent is not. There is a history of technological changes such as Ion PGM™, Ion Proton™, Genexus System, and Ion GeneStudio S5 System.
We have included the following information regarding this comment:
- [page number 7, paragraph 3, lines 261-272]: There is a history of technological changes in Ion Torrent's sequencing platforms, including the Ion PGM, Ion Proton, Ion GeneStudio S5, and Genexus System. Ion PGM was one of the first platforms approved for clinical use and intended for gene panels. It was followed by Ion Proton, which offered higher throughput and extended applications to exomes and transcriptomes. Ion PGM and Ion Proton have been discontinued. Currently, the Ion PGM Dx, an in vitro diagnostic NGS platform based on the Ion PGM, is available. Subsequently, different models of the Ion GeneStudio System (S5, S5 Plus and S5 Prime), a scalable targeted NGS offering a wide range of applications and throughput capabilities, were launched. The Genexus system has recently been launched and is the first NGS solution to incorporate an automated sample-to-report workflow that allows results reports to be generated in a single day (two user touchpoints), presenting potential for clinical application [38].
- [page number 7, paragraph 3, lines 275-286]: However, the Ion PGM and Illumina MiSeq technologies have been compared for their performance in sequencing amplicons for microbiome analysis using various sample types, 16S gene hypervariable regions, and pipelines [39]. Pylro et al. demonstrated that the same biological conclusion was obtained by sequencing the V4 region using both Ion PGM and Illumina MiSeq, employing a stringent quality filter and accurate clustering algorithms [40]. Similarly, Onywera et al. concluded that the cervical microbiome profiles obtained from Ion PGM (V4 region) and the MiSeq (V3-V4 region) were generally comparable [41]. These findings were confirmed by sequencing the V1-V2 region with both platforms from a simulated community of 20 species and in human-derived samples [42]. Loman et al. concluded that MiSeq generated longer reads and lower error rates, while Ion PGM had faster response times [43].
- [page number 7, paragraph 4, lines 287-289]: Finally, Ion PGM has been used to analyze the microbiome in infant fecal samples by sequencing different 16S gene regions, such as V2, V3, V4, and V6, as well as combinations of these, including V3-V4, among others [44].
Comment 6: Line 703: Table 1. Comparison of 16S and metagenomics read length and applications for different sequencing platforms. Illumina: Genus level. The authors state that short-read sequencing methods have resolution up to the genus level, but this is not a clear statement. Based on databases such as SILVA, OTUs or ASVs are determined, and the resolution varies depending on how many species the database contains. And in cases where the evolutionary relationship is very close, such as in the references cited by the authors, databases such as SILVA group the genus name together as Escherichia/Shigella. Another example is the genus Allorhizobium-Neorhizobium-Pararhizobium-Rhizobium. We usually classify species as different if the difference exceeds 3%, but Escherichia and Shigella are 99.7% identical. Can this really mean that the resolution is limited to the genus level due to the limitations of the sequencing platform's technology? The authors are providing incorrect information. It is clearly wrong to say that the special cases where some evolutionarily similar species have differences of no more than 3% but are classified as different genera by scholars are the limitations of the sequencing platform. There is already a paper that states that the 16s rRNA gene has poor resolution for classifying Escherichia and Shigella. Shouldn't this be called the resolution of the 16s rRNA gene?
Thank you very much for your comment, it provides enriching information to the text. We have included the following paragraph:
- [page number 17, paragraph 1, lines 738-741]: It is essential to note that the limitation of resolution down to the genus level of short-read methods cannot only be attributed to the limitations of the sequencing platform, but also to the characteristics of the database used, as well as the taxonomic resolution of the 16S gene itself.
- [page number 17, paragraph 2, lines 742-754]: On one hand, from databases such as SILVA, RDP, Greengenes or NCBI, it is possible to assign sequences to Operational Taxonomic Units (OTUs) or Amplicon Sequence Variants (ASVs). The taxonomic resolution achieved varies depending on the database size (number of taxa) and the resolution capacity (classification level) [133]. In the case of long-read sequencing, a high-resolution database is needed. For example, if 16S gene long-read sequencing results are classified using the SILVA database, which primarily focuses on covering short-read sequences, higher-resolution information may not be fully captured, and in some cases, around 30% of the reads could not be correctly classified [135]. Moreover, taxonomic resolution also depends on the classification tool, and may be higher and faster when using Kraken 2/Bracken rather than tools such as QIIME 2 [134]. Therefore, the choice of sequencing platform is important; however, taxonomic resolution ultimately varies depending on the database and classification tool used. For this reason, depending on the choices, even researchers studying the same topic may obtain different results.
- [page number 17, paragraph 3, lines 758-766]: On the other hand, in cases where the evolutionary relationship is very close, the taxonomic resolution is limited by the 16S gene itself. For example, the 16S gene sequences of the genera Escherichia and Shigella are 99.7% identical [136,137]; however, differences exceeding 3% are typically considered species-specific [138]. For example, in some cases, databases such as SILVA choose to group the genus name as Escherichia/Shigella or Allorhizobium-Neorhizobium-Parhizobium-Rhizobium [139,140]. In these situations, 16S gene taxonomic resolution should be considered for accurate identification.
Therefore, these limitations outside the sequencing platform have to be taken into account, which highlights the complexity of the taxonomic classification process.
We have also changed in Table 1 "Illumina: genus level to mainly genus level".
Comment 7: Line 1101: Table 5. Comparison of the characteristics of commonly targeted regions for taxonomic profiling in microbial community analysis: rRNA operon, full-length 16S rRNA gene and 16S rRNA gene hypervariable regions (V1-V2, V1-V3, V4, V3-V4, V3-V5 and V6-V9). In the case of Illumina, many researchers preferred the V3 region, but it is not introduced in the table. Some researchers have reported that the V4 region has good resolution by improving the traditional universal primer, and some researchers have amplified the V4 region or the V4-V5 region. I would like you to add the V3 and V4-V5 regions as well.
Thank you for your comment. We have included in Table 5 [page number 29, Table 5, line 1167] information regarding the V3 and V4-V5 regions, as well as the reduced bias of the V4 region against the SAR11 bacterial clade with the 806RB primer [216].
Comment 8: About database. The authors' review is very well-intentioned and covers a variety of topics, but it should at least mention that the resolution is higher when using Kraken 2 / Bracken instead of tools like QIIME2, and that the resolution can vary depending on the database or tool used. The sequencing platform is important, but ultimately, the classification varies depending on how dense the database is, so even if researchers study the same subject, the results are different depending on the database they choose. A high-resolution database that matches long-read sequencing is needed. If you classify the actual long-read sequencing results using the SILVA database, the SILVA database only includes efforts to cover short-read sequencing, so it does not cover all the long-read sequencing results with higher resolution, and in some cases, about 30% of the reads are not properly classified. Since changing the sequencing platform alone does not lead to good results when the database is not prepared, please also mention this part.
We appreciate the comment; it contains very appropriate information that complements what is presented in the manuscript. We have therefore included all the information suggested:
- [page number 17, paragraph 2, lines 742-754]: On one hand, from databases such as SILVA, RDP, Greengenes or NCBI, it is possible to assign sequences to Operational Taxonomic Units (OTUs) or Amplicon Sequence Variants (ASVs). The taxonomic resolution achieved varies depending on the database size (number of taxa) and the resolution capacity (classification level) [133]. In the case of long-read sequencing, a high-resolution database is needed. For example, if 16S gene long-read sequencing results are classified using the SILVA database, which primarily focuses on covering short-read sequences, higher-resolution information may not be fully captured, and in some cases, around 30% of the reads could not be correctly classified [135]. Moreover, taxonomic resolution also depends on the classification tool, and may be higher and faster when using Kraken 2/Bracken rather than tools such as QIIME 2 [134]. Therefore, the choice of sequencing platform is important; however, taxonomic resolution ultimately varies depending on the database and classification tool used. For this reason, depending on the choices, even researchers studying the same topic may obtain different results.

Reviewer 2 Report
Comments and Suggestions for Authors
This is a long and thorough review of metagenomic and microbiome sequencing technologies and protocols. The writing is clear and engaging and well researched. I've worked in managerial/executive positions in 2 large sequencing companies, both were platform agnostic, so I have a good insight into the benefits of all sequencing technologies and approaches. The big problem with the manuscript is that the authors don't address the most important aspect of sequencing. The cost per Gb of a set quality of sequence data. The cost of the instrument is not particularly important in many cases, but the cost of reagents to obtain a Gb of data of say average Q30 is very important. The dismal science (economics) is unfortunately important even for scientists. The authors discussion of long reads and the benefits of Nanopore all make a lot of sense and are routinely demonstrated in the literature. Unravelling the actual cost of doing a particular project with the different platforms is difficult, mainly as the supplying companies spend a lot of time obfuscating what the cost is. However, it can be done, and often Illumina is the cheapest. Not always, as minimum sequence data for sequencing a library are determined by flow cell sizes, instrument, and other factors.
Why are long reads good? The authors say it's good for making longer contigs with metagenomic sequencing. True. And also true (and more important) is longer reads means more accuracy both at species and strain level. They don't really mention with any depth that sequencing depth determines sensitivity (what the threshold of detection of a bug is in a mix). This sequencing depth comes at a cost, and usually this means Illumina for metagenomics. For amplicon sequencing often Nanopore is a cheaper alternative, though this also depends on the diversity of organisms and their abundances across a log scale of concentrations. This is all critical information for comparison between the platforms. In my sequencing companies, we sold and used all platforms as they all do have advantages in certain circumstances. Mostly though, Illumina and Nanopore. Much of what is useful to a customer is a non biased recommendation to use a particular platform. Unfortunately what's the best approach is messy and complex. Its mostly about how much money the customer has, unfortunately.
line 22 - long read
line 25 - no mention of cost criteria, ouch!
line 120 - not sure what 3rd one is?
line 394 - pacbio and Nanopore (not NGS?)
line 1150 - which it comes from (actually this is rarely true. Most reads easily assign to an organism.
Genome of a community? Do you mean the pan-genome of a community or the metagenome of a community?
line 1183 - read depth determines sensitivity (detect rarer bugs). Genome assembly from metagenomes isn't that common, though yes. Novel genomes require more depth to assemble (sequence can be inferred for known bugs).
line 1187. Read depth mainly helps sensitivity rather than taxon specificity. This conclusion is very dependent on the type of sample. The complexity of tropical soil would require much more depth compared to western fecal samples.
The authors have done great work summarising, unfortunately they've avoided the dismal science, which I can very much understand, but in the case of comparison of sequencing platforms, its a grim focal point.
Author Response
Thank you. We have uploaded an attachment.

Round 2
Reviewer 1 Report
Comments and Suggestions for Authors
The authors have faithfully reflected the contents of the first review and revised the manuscript by referring to numerous references. I think it must have taken a lot of time, but I appreciate the authors' efforts and I have no further suggestions for the revised manuscript.
Author Response
Thank you very much! The comments have greatly enriched the work.